

# Evaluation of Economic Impacts from Flood Damages Using Hybrid Input-Output Analysis

Cholapat JONGDEEPAISAL, Kohei YOSHIMURA, Seigo NASU

School of Economics and Management, Kochi University of Technology, 2-22 Eikokuji, Kochi City, Kochi, Japan

Correspondence to: Cholapat Jongdeepaisal (cholapatlay@gmail.com)

**Abstract.** Flooding is a common issue in many countries and can cause extensive damage to residential areas, agricultural and livestock areas, and public facilities. These damages impact the local economy directly and create unexpected demand for goods and services. The consequences of flood damage are marked by the ambiguity of how it changes the production of local economies. For this reason, our study implements the use of the hybrid I-O analysis to evaluate the economic effects of flood

damages. The flooding scenario in Kochi prefecture, Japan simulated from the inundation model was selected to demonstrate the analysis. The influences of flood damages were divided into two stages, the garbage cleaning stage and the reconstructing stage; these two stages lead to the different scenarios based on the different activities. It was found that in total, flood damages could stimulate economic growth when the positive effect of reconstruction activities surpass the negative effect of garbage cleaning activities.

## 1 Introduction

The increase in precipitation driven by global warming has risen the awareness of people on a global scale. The inevitable consequences of the greater frequency and magnitude of flooding have been experienced in many countries [Mahmood et al, 2016, Koks et al., 2016]. The damages from floods subsequently affect economic growth [Skidmore et al., 2002]. This has not only occurred in the tropical zone countries; Kochi prefecture, Japan has also experienced and suffered from high annual

rainfall as a result of flooding in some urban and agriculture areas [Shikoku Disaster Information Archives]. The emerging costs from these incidents fall upon the local residences themselves and support from the government. Some prior studies have considered the cost of maintaining the damaged equipment and facilities as an economic loss or a loss of capital, while some studies have indicated that this cost could be considered a brutal stimulant to economic growth [Haraguchi et al., 2015, Mendoza-Tinoco et al., 2017, Grames et al., 2016]. Many studies may have overlooked the presence of the cost of removing

garbage and space cleaning, which is considered significant to assess the total impacts of these disasters.

Concerning these arguments, our study attempts to identify the flood damages through a new economic sector (garbage cleaning sector), which is the input of the hybrid I-O analysis, to identify the economic impacts of flood damages on the local economy. The hybrid I-O model starts by collecting data forecasted by the 2-d inundation model. The affected area with a high





probability of flooding is then identified. This area is nearby Kochi city, where the facilities are houses, shops, and some industries. This study mainly focuses on analysing the effects of flood on the general asset damages on buildings, furniture, electrical appliances, and shops, and companies. The flood damages estimation method from the Flood Control Survey Manual [Flood Control Economic Survey Manual, 2005], which was made available by Japan's Ministry of Land, Infrastructure, and

Transportation database, was utilized for the damage assessment.

Flood damages, in this article, refer to general asset damages which include damage to buildings, furniture, electrical appliances, shops, and company equipment in a residential area, industrial area, or public facilities and utilities. The newly conceptualised framework of this study is that the flood damages are described by two remediated activities, garbage cleaning,

and reconstruction. The garbage cleaning activities including removing garbage and debris, maintaining the access roads, and constructing temporary facilities, occur during the short period after the local area is damaged by a flood. The reconstruction activities including maintaining all buildings and facilities in proper condition and restoring the damaged furniture and electrical appliances takes place over a long period of time after the rehabilitation activities. The costs arising from these activities are referred to as garbage cleaning costs and reconstruction costs, respectively. These costs will reflect the total

impact of flood damages on the local economy by addressing each kind of cost in the economic sector and will be analysed through the hybrid I-O analysis.

Unlike ordinary I-O analysis, the hybrid I-O analysis has an extension that can create a connection between a new industry and the local economy and allocate the change of transferred resources and products. The new industry certainly consumes

some resources from the economy, while the industry itself produces its products and supplies them back to the economy. Regarding the hybrid I-O analysis, the resources and products transfer among the new industry and the economy are termed cut-off resources and cut-off products. The cut-off term is defined by the process of obtaining the resource that once was used by the other economic sectors or industries. Supposing that the availability of the resource in the economy is limited, the rehabilitation of the damaged area or garbage cleaning sector competitively procures some resources and finally obtains them,

while some other economic sectors lose their resources. The resources are then transferred from the economy to the garbage cleaning sector for garbage cleaning activities. The products of the garbage cleaning sector are certain to aid the public demand for garbage cleaning. Once the garbage cleaning activities are finished, the reconstruction activities are the next step to return the damaged area to the operability condition, such as building and structure reconstruction, houses and working space refurbishment, and electrical equipment and appliances maintenance. These requirements raise the demand for goods and

services from some specific sectors and could be of benefit to those sectors. Thus far, these cause and effect scenarios could be investigated using the hybrid I-O analysis, but how to implement this is the key challenge of this study.



## 2 Methodology

### 2.1 Conventional Input-Output Analysis

Input-output (I-O) analysis is an economic tool for assessing and predicting impacts such as the change in demand for goods and services of either the economic sectors or other forms of consumption, in terms of the change regarding the future

economy's structure on a macro-scale [Leontief, 1966, 2012, Miller et al., 2009]. The conventional I-O analysis is expressed by the following equation:

$$A = Z/X$$
$$X = XA + f \qquad\qquad 1$$
$$X = (I - A)^{-1} f$$

where the technical coefficient ($A$) represents the ratio of resource trading among the economic sectors generated by dividing the transaction matrix ($Z$) by the total production ($X$). $I$ is the identity matrix of $A$ and final demand ($f$) is the other consumptions that are not included in the economic sectors, such as household demand, government demand, other private investment

demand, and export and import demand. Rewriting the equation gives the basic Leontief's function, where $X$ is proportional to the change of demand $f$.

The I-O method has been developed and implemented through several decades from the fundamental use to applied approach to some specific analyses such as water footprint, ecological footprint, $CO_2$ and $SO_2$ emissions, and energy embodied [Cláudia

et al., 2014, Liu et al., 2015, Onat et al., 2014, Guill et al., 2015, Chen et al., 2015]. Another highlighted method that constitutes an integration of the industry's processes and the economy is the hybrid of life-cycle assessment (LCA) and I-O analysis [Rocco et al., 2016, Nagashima et al., 2016]. This significant hybrid approach between LCA and I-O leads to the new hybrid model of physical and monetary units.

2.2 Hybrid Input-Output Analysis Methodology

The hybrid I-O analysis presented in this study is the hybrid of the physical unit and monetary unit inside the I-O table. This method creates a new virtual industry inside the existing I-O table as well as the consumption and production relationship between the new industry and the local economy [Jongdeepaisal et al., 2018]. The structure of the hybrid I-O analysis is described by the following equations:

$$\begin{bmatrix} X_P \\ X_M \end{bmatrix} = \begin{bmatrix} P & C_d \\ C_u & M \end{bmatrix} + \begin{bmatrix} f_P \\ f_M \end{bmatrix} \qquad\qquad 2$$

where physical submatrix (P) refers to the new introducing industry, in which each extensive row and column are the

consuming and producing processes of the new industry. The upstream cut-off submatrix (Cu) and downstream cut-off submatrix (Cd) present the transferred resources from the economy to the new industry and from the new industry to the economy, respectively. Monetary submatrix (M) refers to the Kochi prefecture I-O table or the local economy where the flood occurs. P and Cd are presented in the physical unit and the Cu and M are addressed by the monetary unit.





The cut-off method is introduced to form the relationship between P, Cu, Cd, and M submatrices. This method is used for transferring the resources or products from the economic sectors to the new industry. Each economic sector has to give away
their resource to the new industry equivalent to the cut-off portion Cp determined by:

$$C_p = A_{ij} \times C_t$$

where Aij is the technical coefficient and Ct is the total cut-off portion of the total resource requirement of the new industry. Thus, the Ct is the transferred resource from M to Cu, and the economic sectors in M reduce their consumption equivalent to Cp. Two terms must be denoted here are 'cut-off resources' and 'cut-off products'. The cut-off resources are the resources that prior belong to the economic sectors in the economy (M), but they are cut-off to Cu as the consumption required by the new
industry. The cut-off products are the products that transfer from M to Cd in an equivalent amount to the new industrial production amount. This cut-off is necessary for maintaining the demand for products of the new industry and the existing industry. The new industry may induce an increasing economic sector's demand, but it does not directly change instantly. If we imagine that the demand for resources in the economy is limited at one point of time and the new industry can compete in the sale of their product with the existing industry, the existing industry in response loses demand for the product and
subsequently reduces their productivity.

## 3 Flood Damage Classification and Allocation

The GIS data represented by a 50x50 square meter grid, which includes the land use, ground height above sea level, and other geographical data, was made available by Japan's government. From this data, the 2d-inundation model was used to simulate the flood area and flood water level. From the results of the 2d-inundation model, the flood occupied the southern area of the
Kochi city, in which residences, shops, companies, and agricultural areas are located. Once the flood area is identified, the calculation of flood damages is executed by analysing each land use inside the grid map such as houses, shops and companies, public facilities, and agriculture lands. The flood damages are then calculated based on the Flood Control Economic Survey Manual applicable in Japan's Ministry of Land, Infrastructure, Transport, and Tourism database. The total flood damages are categorised into each type of damage including building structure damage, furniture, and household appliance damage,
industrial equipment damage, industrial product damage, shop and company equipment damage, shop and company product damage, agricultural machine damage, agricultural product damage, and public facility damage (Table 1).






**Table 1. Category of General Asset Damages Induced by Flooding**

| Categorized Flood Damage | Yen (¥) |
| --- | --- |
| Building structure damage | 4,304,243,032 |
| Furniture and household appliance damage | 3,995,107,489 |
| Industrial equipment damage | 514,706,278 |
| Industrial products damage | 432,493,570 |
| Shop and company equipment damage | 1,094,693,972 |
| Shop and company product damage | 140,691,963 |
| Agricultural machine damage | 1,729,605 |
| Agricultural product damage | 774,283 |
| Public facility damage | 7,810,907,942 |
| Total Flood damages | 18,295,348,133 |

From the total flood damages, there are two types of costs including the reconstruction cost and garbage cleaning cost. The cost of reconstructing all facilities is expected to be equivalent to the total flood damages. The expenditure of 18,295,348,133

5 Yen in total for flood damages has to be paid to purchase goods and services for resuming the condition of the local area. Additionally, each type of damage is then further classified into the related economic sectors; for example, building structure damage is allocated to the purchase of construction and real estate sector for reconstructing purpose. Thus, the resource's demand for reconstruction is allocated into the related sectors based on a damage type (Fig.1).

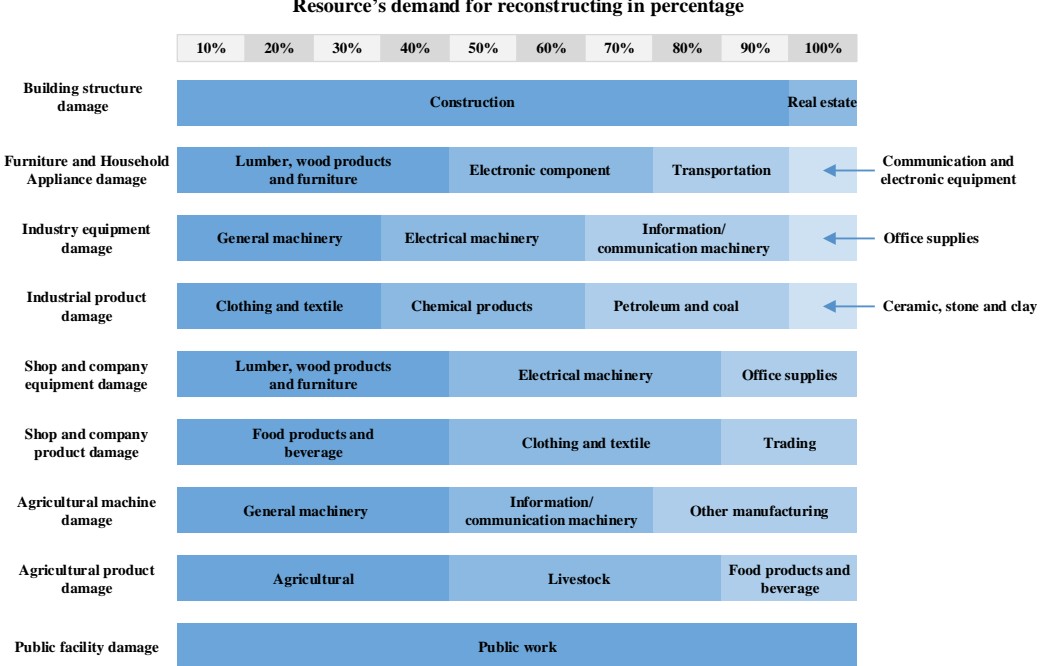

10 **Fig. 1. Resource Demand for Reconstruction from the Relevant Economic Sectors as a Percentage**



In addition, the cost of garbage cleaning works is estimated based on the total flood damages, about 8% of the total flood damages. This cost is spent on manpower and trucks that are necessary for clearing obstacles and cleaning garbage. The garbage cleaning cost emerges in an early stage soon after the flood struck the damaged area and the reconstruction cost appears thereafter. Therefore, the calculation will be presented in two steps and each step shows how each cost for garbage cleaning and reconstruction can be evaluated using hybrid input-output analysis.

## 4 Structure of Hybrid I-O Model for Flood Damages

### 4.1 Garbage Cleaning Sector in Physical Submatrix

It might be intriguing that we imagine the new activities that appeared from the flood damages as a new industry. It is a fact that floods destroy buildings, facilities, household furniture, and electrical appliances, resulting in additional demand for goods and services for garbage cleaning and reconstructing the damaged area [Li et al., 2013, Hallegatte, 2008]. Since the garbage cleaning service and reconstruction for flood damages do not exist in the existing economy, the new process of consuming resources and producing the product has to be set up in the I-O table. A virtual sector called the garbage cleaning sector is then added to the existing economy in response to the new activities. This sector consists of three processes including truck and other transportation use, manpower use, and garbage cleaning service. Trucks and manpower are the main resources which already existed in the local economy. If the resources in the local economy are limited, these two resources are cut-off from the local economy to serve the demand for garbage cleaning activities. The truck is cut-off from the transportation sector directly. However, there is no manpower sector in the existing economy so we must create a new manpower sector especially for this purpose (section 4.2). The assumption has been made that the manpower required for the garbage cleaning activities is only the construction manpower which is likely to perform similar works. The manpower is then cut-off from the manpower row, where the construction sector is previously consumed. The garbage cleaning sector produces the garbage cleaning service as a product and this product is provided in return to the local economy or the main sector that is affected by flood damage including the agricultural sector, public work sector, and household and government demand.

### 4.2 Manpower Row and Column Setting

The garbage cleaning activities likely require only manpower, excluding the materials or goods to accomplish the work. Regarding the I-O table, the manpower sector's consumptions and products prior exist in the value-added and final demand. Thus, we intend to migrate this manpower to the inter-industry region as the manpower sector to assess the impact of the manpower used for garbage cleaning activities (Table 2).

30





**Table 2. Manpower Sector inside the I-O Table**

| I-O Table | S | MP | f | X |
|---|---|---|---|---|
| S | $Z_{11}$ | $Z_{12}$ | $f_1$ | $X_1$ |
| MP | $Z_{21}$ | $Z_{22}$ | $f_2$ | $X_2$ |
| VA | $VA_1$ | $VA_2$ | | |
| X | $X_1$ | $X_2$ | | |

The economy's I-O table consists of the economic sectors (S), manpower sector (MP), final demand (f), and value-added (VA), where the total production (X) is the summation of each vector of input and output. The manpower is separated from value-added directly to create a new row of the manpower sector. For the manpower sector column, we divide it from the final demand column by the following equation:

$$X_1 = Z_{11} + Z_{12} + f_1$$
$$X_1 = Z_{11} + Z_{21} + VA_1$$

where $X_1$ is the economic sector's total production, $Z_{11}$ is the economic sector's goods that consumed by the economic sector, $Z_{12}$ is the economic sector's goods that consumed by the manpower sector, $f_1$ is the economic sector's final demand, $Z_{21}$ is the manpower sector's goods that consumed by the economic sector, and $VA_1$ is the value-added that consumed by the economic sector. By solving the above equation, the manpower sector's column is expressed by the following equation:

$$Z_{12} = Z_{21} + VA_1 - f_1$$

The manpower products that once belonged to the construction sector are now cut-off as the garbage cleaning demand. As a result, some negative and positive values present in the manpower consumption consuming column (Table 3).

**Table 3. Part of the Column of Input to the Manpower Sector**

| Economic Sector | Manpower Sector (M¥) |
|---|---|
| Agricultural | -16,997 |
| Livestock | 5,075 |
| Forestry | 3,690 |
| Fishery | -7,026 |
| Mining | 4,289 |
| Food and Beverage | -26,403 |
| Textile | 6,732 |
| Lumber, Wood Product and Furniture | 6,664 |
| Pulp, Paper, and Paper Product | -4,680 |
| Chemical Product | 78,374 |
| Petroleum and Coal | 59,616 |
| Plastic and Rubber | 24,782 |



### 4.3 Calculating Procedure of Hybrid I-O Analysis

The calculation procedure inside hybrid I-O analysis is separated into pre-process, intermediate-process, and post-process (Fig.2). The pre-process is the first step for changing the economy's structure in response to the garbage cleaning activities including constructing P, Cu, Cd, and M, and utilizing a cut-off method to transfer the resources and products from M to Cu
and Cd. The garbage cleaning sector purchases of goods and services from the economy for garbage cleaning purpose. This action could be determined by the cut-off resources, where the resources in M are transferred to Cu. Then, the cost of removing the garbage and clearing space is responsible for the main affected sectors such as the agricultural sector and public work, and non-sectors such as the local government and households. It is certain that the government, private sector, and households unavoidably have to purchase the goods and services for resuming the condition of the city; otherwise, people could not
continue their daily life. Therefore, the resources are purchased and utilised by the garbage cleaning sector for their activities, while the burden cost is left to be paid by the affected economic sector and the final demand.

The intermediate process is to maintain the technical coefficient value after cut-offs. When the cut-off was made from M to Cu and Cd, the resources that input to the economic sectors are reduced equivalent to the cut-off portion and transfer to garbage
cleaning input. The remaining input to the economic sectors affects the change of technical coefficient value. The technical coefficient represents the trade ratio from one sector to another or it is the structure of the economy. The technical coefficient cannot be changed rapidly, but it is gradually changed due to technological advancement. To do this, we firstly separate M from the other submatrices. Then, we detach the row and column of the sector that is cut-off their products to Cd. Thus, this sector will not affect by the following process. Regarding the cut-off resources to Cu, the loss in the economic sector's input
induce a loss in the total production. To maintain the existing economic structure, the technical coefficient before the cut-off resources is used to recalculate with the reduced total production. After that, we reattach the row and column of the cut-off product sector. As a result, the loss through the economic sector regarding the necessary resources for garbage cleaning activities appears in the loss in total production.

Finally, the post-process determines the rise of demand for additional goods and services initiated by the flood damages. The reconstruction cost is used to purchase goods and services to resume the good condition of building structure, furniture, and household appliance, industrial equipment, and shop and company equipment to the operational condition. This expenditure is considered as the economic stimulant. For instance, Kochi prefecture residents buy their new furniture and appliances and repair their houses and garages. Private companies and shops also refurbish and redecorate the working space, while the
government has to invest in maintaining all public facilities such as roads, bridges, and other utilities. Therefore, the additional demand for reconstructing the facilities is placed on the related economic sector's final demand, which results in a benefit to the local economy [Okuyama, 2007].





**Pre-Process**

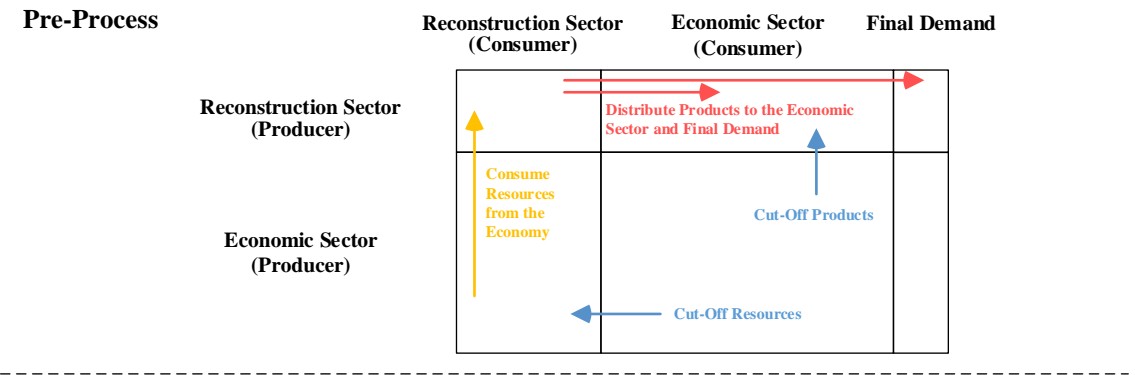

**Intermediate-Process**

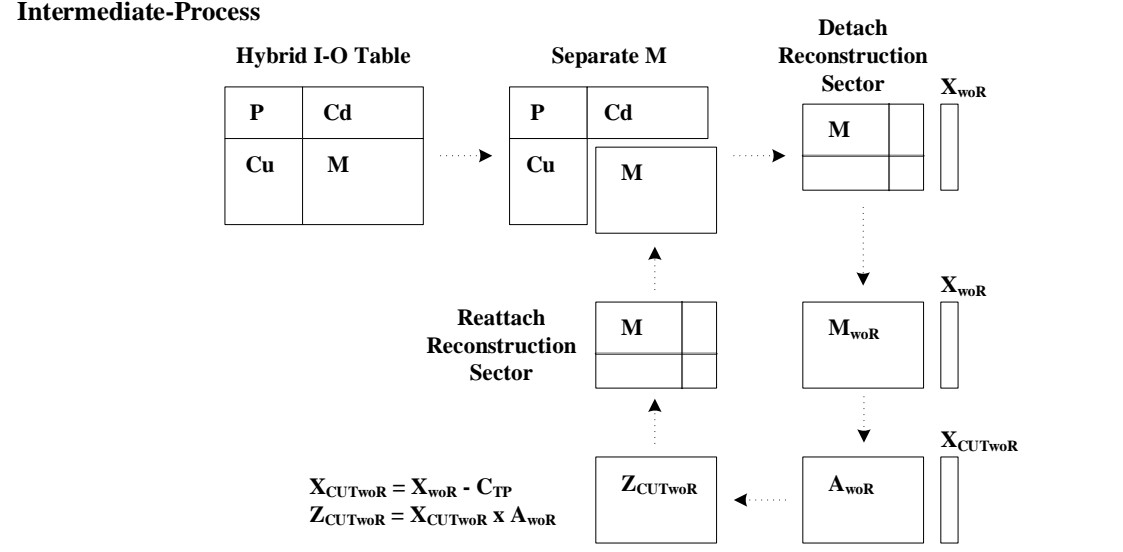

**Post-Process**

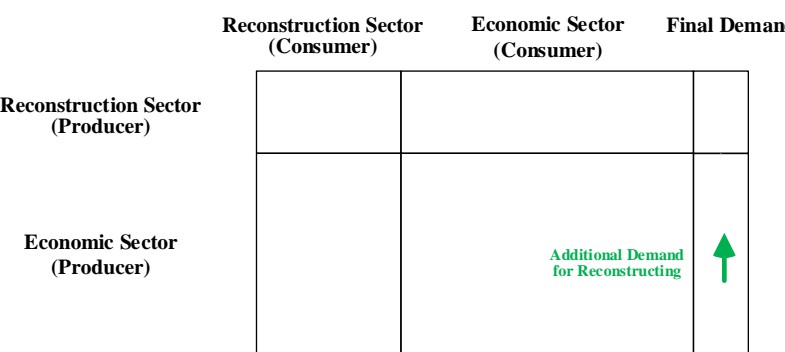

**Fig. 2. Calculation Procedure of Hybrid I-O Analysis for Evaluating the Flood Damages**



## 5 Calculation Results and Discussion

The results from the calculation are presented in two steps: [1] the negative economic effect of garbage cleaning cost in the early stage, and [2] the total economic effect when the reconstruction cost is included, and this cost is considered a benefit to stimulate the growth of the local economy.

### 5.1 Negative Effect from Garbage Cleaning Activities

Flood damages initiate the loss in the economy's total production by acquiring the goods and services from manpower and transportation sectors for garbage cleaning. The garbage cleaning cost appears in the early stage of flood and must be paid by some economic sectors, households, and the government. Consequently, the demand for goods and services of some economic sectors temporarily decreases to meet the requirement for garbage cleaning activities. Regarding the calculation process, the

technical coefficient pre-adjustment method is applied to firstly conserve the existing resource distribution ratio by keeping the technical coefficient constant and secondly to examine the economic effect of resource shifting to the garbage cleaning activities. As a result of using the technical coefficient pre-adjustment method, the influence of the cut-off resources from the manpower and transportation sectors to the garbage cleaning sector initiates other subsidiary losses throughout the other economic sectors. This indirect economic loss is crucial to reveal the complexity of the impact of flood damages [Zhang et al,

2017].

Fig. 3 illustrates the reduction of the total production of each economic sector compared to the existing total production of Kochi prefecture, in Japanese Yen. The new garbage cleaning sector generates the product sale of garbage cleaning services of about +1,381 million Yen to the Kochi's local economy. Many sectors likely suffer from the loss in resource consumption

of manpower and transportation sectors. If the sectors lose their consumption of resources, they also produce fewer products to sustain the economy demand. Therefore, it certainly results in the reduction of products of the related economic sectors. The construction sector is the top loser of about -1,371 million Yen because the demand for construction is transferred to the garbage cleaning services to perform the activities. The directly related sectors, the manpower sector and the transportation sector, lose -741 and -188 million Yen, respectively. The other economic sectors suffer from the indirect impacts, which are

initiated by the garbage cleaning activities, including business services (-324 million Yen), trading (-172 million Yen), metal product (-149 million Yen), petroleum and coal (-163 million Yen), and lumber, wood products and furniture (-87 million Yen). In total, the economy loses about -2,056 million Yen for garbage cleaning purposes. When the resource demand for producing the product of these sectors shrinks or is cut-off to the garbage cleaning sector, it simultaneously initiates the loss of the other economic sectors' resource demand for production.






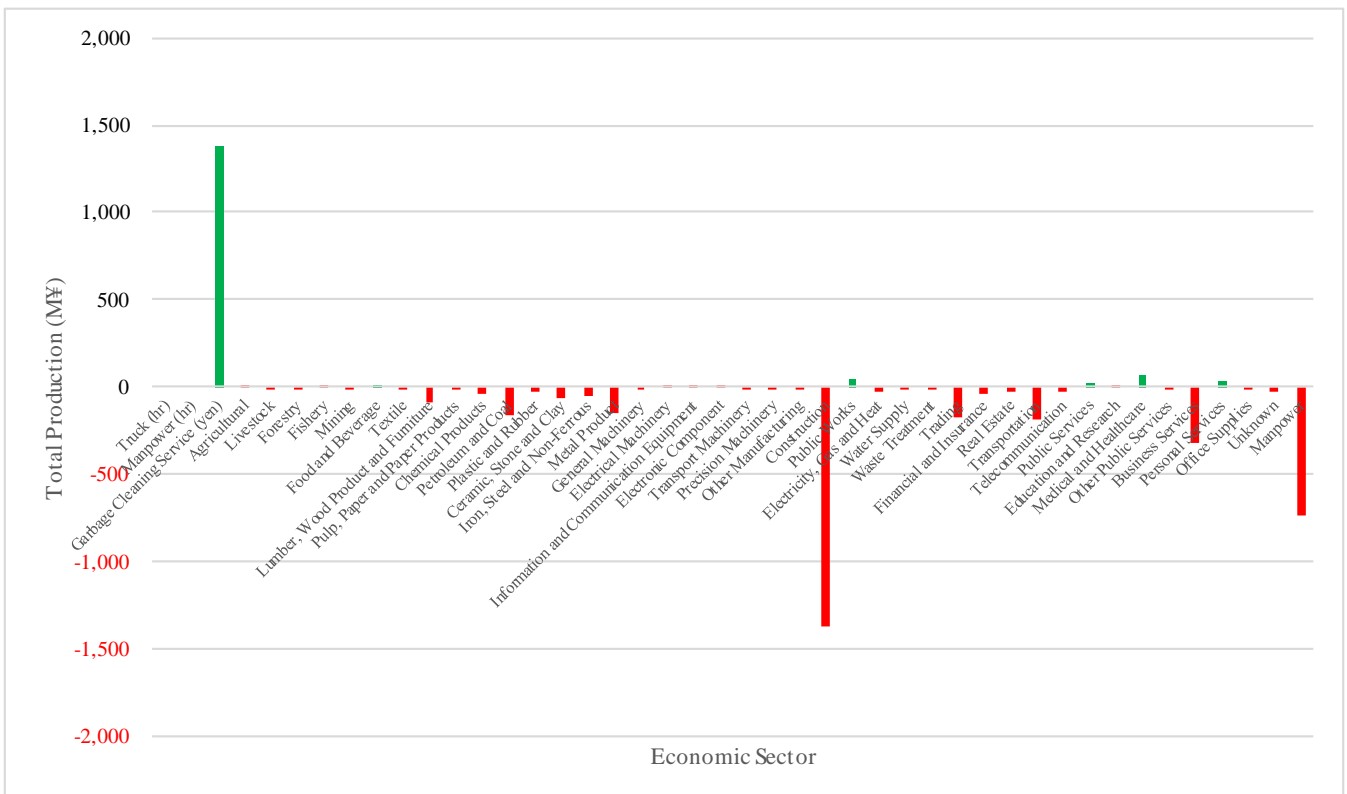

**Fig. 3. The Economic Effect from Garbage Cleaning Activities by Comparing the Total Production Before and After Technical Coefficient Pre-Adjustment Method in Japanese Yen**

### 5.2 Total Economic Impact Influenced by the Flood Damages

5  Flooding destroys houses, shops and companies, agriculture areas, public facilities, and the surrounding landscape. The reconstruction cost must be spent to recover the damage from the flood. This cost is considered to be the factor for vitalising the economy because it initiates increasing demand for goods and services from some economic sectors. Households, private investment and the government have to spend their money on maintenance and renew their facilities. Regarding the calculation, the additional demand for goods and services is allocated to final demand in the hybrid I-O table.

Fig. 4 displays the total production of each economic sector influenced by the effect of both garbage cleaning and reconstructing activities by comparing with the existing total production of Kochi prefecture, in Japanese Yen. It appears that the manpower sector gains the highest benefit of about +9,858 million Yen in total production. Public work and business services also obtain significant gains at about +6,948 and +4,475 million Yen of total production respectively. Lumber, wood

15  products and furniture, petroleum and coal, iron, steel and non-ferrous, precision machinery, construction, trading, and transportation production also increase at least +2,000 million Yen.





There is a major loss in the medical and healthcare sector of about -1,458 million Yen in total production, while personal services, public services, food and beverage, agricultural, and fishery also suffer minor losses.

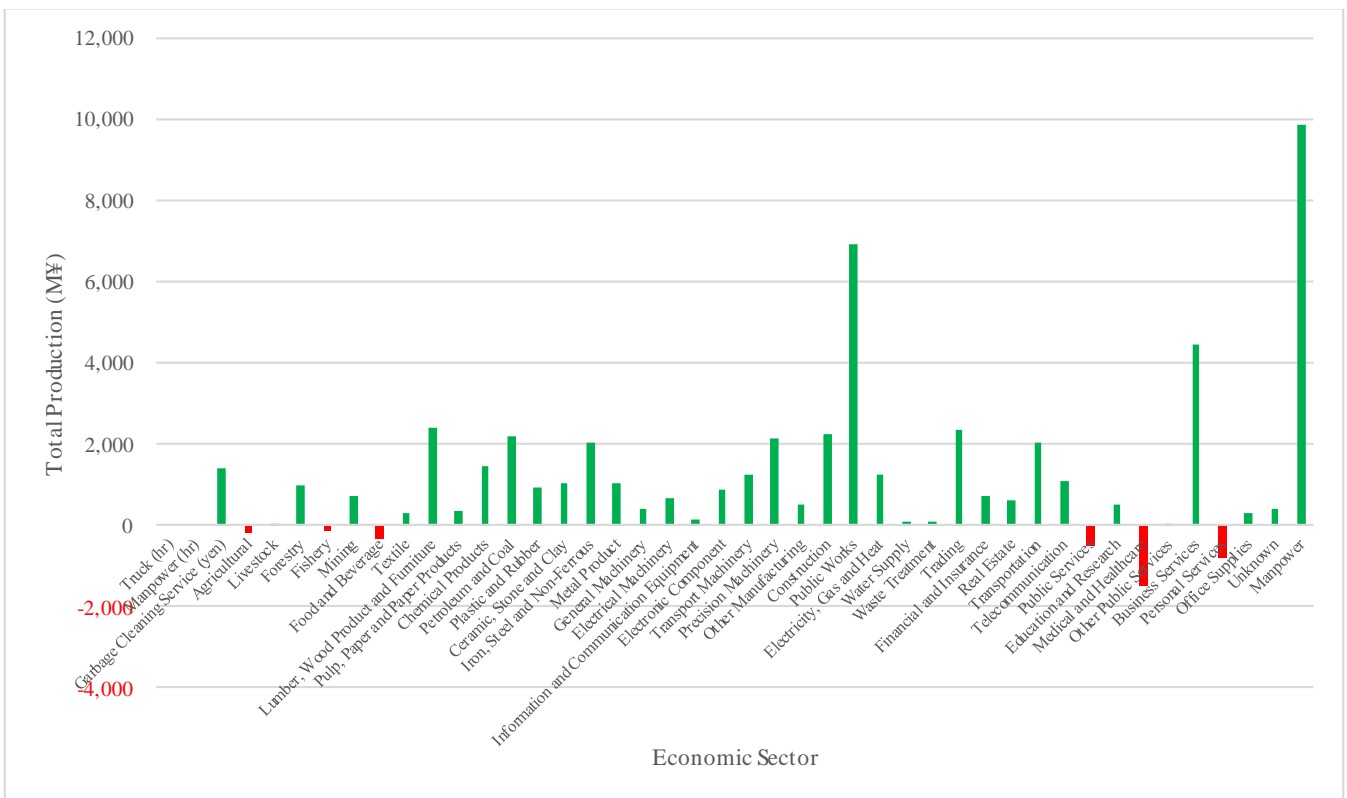

**Fig. 4. The Economic Effect of Reconstruction by Comparing the Total Production Before and After Adding Demand for**
5  **Reconstruction in Japanese Yen**

### 5.3 Discussion

The garbage cleaning activities acquire the manpower and trucks from the economic sectors to perform the cleaning services. The resources taken away by these activities initiate a loss in the related economic sectors, which results in a shrink of the economy's total production. On the contrary, the additional demand for reconstructing the damaged facilities mobilizes the
10  growth of the economy. Many goods and services are needed by private investors, households, and governments, which consequently increases the total production of related economic sectors. In summary, the flood damages inflict direct loss from garbage cleaning activities, resulting in -2,056 million Yen of depletion of the economy's total production, while the reconstruction cost stimulates economic growth by +52,670 million Yen. The total economy's production increases by 50,614 million Yen and accounted for 1.024% of the overall total production. Damages are an unfortunate loss, but from an economic
15  productivity perspective, it is also an opportunity for economic growth.



The use of hybrid I-O analysis leads to the new model adjustment as well as the new finding of economic impact characteristics from flood damages. The manpower sector is newly created in response to the demand for garbage cleaning services, which is added as an extended economic sector. In value-added, the manpower exists in the form of the income for labor that each sector has to pay to produce goods. This income is later paid back to the economy as demand for goods and services for

household living, which is allocated in the final demand. In the hybrid I-O table, the manpower sector's row is cut-off directly equivalent to the labor compensation in value-added, while the manpower sector's column is cut-off from the final demand following equation 5. Noticeably, some values in the manpower sector's column are found negative. A negative value in the I-O table signifies a byproduct and it is also similar to the negative value presented in the manpower sector's column. If one observes the definition of negative value in the manpower sector's row, the negative value means that the manpower sector

did not consume the products, but it produces the product instead. Referring to equation 5, if the final demand at a specific row is higher than the value-added in the column of the same sector, the value in the manpower's column becomes negative. If we recognise $Z_{21}+VA_1$ as a value-added and $f_1$ as demand for that specific sector, $Z_{21}+VA_1<f_1$ means that people consume the products more than their capability of producing value-added. Each industry has positive and negative values in the manpower column; therefore, the total amount of the manpower column is zero. In case there is a negative value at the specific sector in

the ordinary I-O table, it is a byproduct. In the same case of the hybrid I-O table, practically it is a negative demand, which has the same effect of by-product to fulfill the final demand. It means that a large number of $f_1$ is allowed by the negative value in this specific sector.

Public work, business services, and manpower sectors are benefited the most from the additional demand for reconstruction.

The public work sector gains this benefit directly through increasing demand for reconstructing the public facilities. On the other hand, the manpower sector and business service sector are induced by indirect demand for goods and services from the other economic sector's consumption. The products of these two sectors are the major requirement for the other economic sector, so an increase in demand for reconstruction significantly impacts the additional number of products required in the economy, and thus indirectly induces the manpower sector and business service sector.


The structure of the hybrid I-O analysis also allows one to trace the resource use in each process. The resources from economy M to Cu for the garbage cleaning work, the process of utilizing resources of the garbage cleaning sector, as well as the product of the garbage cleaning sector that is fed back to the economy in Cd, can be visualised. Moreover, by the hybrid I-O method, the resources transfer from the economy to use in the new sector's process can be analyzed in both the physical unit and

monetary unit. This advantage of unit transition leads to the finding of implementation on a unit price adjustment. When the resource is cut-off from M and presents in Cu, the unit price of each resource can be individually adjusted regardless of the prior unit price. This price can represent the new price of the resource that the new sector purchases and further uses for their process of generating products. It is yet to be present in this study, but it is likely that the unit price for garbage cleaning cost



is not equivalent to the existing price purchased by the prior sector. In that case, the advantage of hybrid I-O analysis of unit price adjustment could play an important role in assessing the flood damages.

## 6 Conclusions

In conclusion, this study presents the new concept of using the hybrid I-O analysis to evaluate the economic impacts of flood damages in the form of the new garbage cleaning sector. The flood damages are described through two new scenarios, the garbage cleaning stage and the reconstructing stage, in which each scenario appears in a different period. The cost of garbage cleaning and reconstruction are recognised as significant factors to assess the economic impacts of flood damages. The flood damages stimulate economic growth where the positive effect of reconstruction activities is overthrown by the negative effect of garbage cleaning activities. The benefits to the local economy's total production are reflected by the additional total production of each economic sector. The study proves the usefulness of hybrid I-O analysis, not only for assessing the economic sector, but also for adapting the analysis to other scenarios such as flooding. The use of the model could enhance the damage prediction and also enable the government and policymakers to find alternative solutions for mitigating damages.

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
