# Peer review of "Evaluation of Economic Impacts from Flood Damages Using Hybrid Input-Output Analysis"

_Natural Hazards and Earth System Sciences, 2020_

## Referee Comment (RC1) · Anonymous Referee #1 · 7 Jan 2021

Thank you for submitting this article. It is very well-written and presented in a very clear manner. The focus on the garbage cleaning and reconstruction costs is important.

The main question I have is whether there is any need to develop new virtual sectors. I would hesitate to make the statement "Since the garbage cleaning service and reconstruction for flood damages do not exist in the existing economy, the new process of consuming resources and producing the product has to be set up in the I-O table" (line 10). I find this hard to comprehend conceptually. Surely the cleaning and construction industry already exists prior to the disaster and you're merely employing more resources to rebuild? As such, in accounting for the next IO table, these extra

"production" from flood damages would be included.

Some of the technical comments are outlined below.

Line 21, it would be good to add role of international organisations such as the World Bank.

Line 24-25: please cite the studies that have looked at the cost of removing garbage and space cleaning. What is the evidence on the cost of removing garbage in percentages – any previous studies?

Lines 18-31: might be a good idea to add a diagram/figure to illustrate the difference between traditional IO analysis and hybrid IO.

---

## Referee Comment (RC2) · Anonymous Referee #2 · 7 Jan 2021

In this paper, the economic impacts of flooding are simulated with the so-called hybrid IO method. The description of the methodology in Section 2 is incomplete and in contraction with the description of the actual assumptions made in Section 3 and 4. Moreover, the application of method suffers from several unrealistic assumptions. Hence, a total rewrite is necessary. In detail: (1) Equation 2 is incomplete. What is needed is the specification of the full IO table with the two virtual industries (garbage cleaning and reconstruction), which is only partially done in Table 2. In Section 4 and Table 2 it is explained rightly that garbage cleaning inter alia needs manpower, which is taken from the manpower row of existing industries. Amongst others, this is absent in (2). (2) The cut-off method for resources described on p.3 is too rude. Equation 3

implies that structure of the use of resources from the rest of the economy by the new industry Cu is equal to the structure of the aggregate use of those resources in the rest of the economy Mi, where i is a summation vector with ones. This a highly unrealistic assumption because the new virtual industries take care of garbage cleaning and reconstruction, which both have cost structures that will be quite different from the cost structure of the rest of the Kochi economy that is dominated by services. In fact, Section 3, describing the details of the procedure for garbage cleaning and reconstruction activities, is contradictory to (3), but more realistic. (3) The cut-off method for products also described on p.3 is imprecise and not convincing. Equation 4 that should describe the structure of Cd is lacking. The most plausible solution would be to assume that the new industries do not deliver intermediate products (i.e. Cd is zero), but only deliver final products to fP, which goes at the cost of the final demand for products from the rest of the economy fM. (4) As a consequence of the inconsistency of equation 5 (see Major details) Table 3 delivers nonsense. (5) can only be used to establish the overall total of z12 (i.e. a small case as it is not a matrix, but a column). Instead of Table 3, the structure of the inputs to the manpower sector should resemble the structure of the household consumption part of the final demand column (see Oosterhaven, Rethinking IO Analysis, Springer 2019, ch. 4). The procedure that describes from which sectors the manpower of garbage cleaning is taken should be specified independently. It is not related with the structure of household consumption demand.

Major details: • P.1-2 & p.12-14. This article does not mention the direct los of output of industries that have to partially or entirely close down due to the simulated flood nor does it mention the indirect impacts of these close down in the rest of the economy nor does it mention the cost in the rest of the economy of raising the money for the cleaning and reconstruction activities. The article, consequently, presents a far too optimistic view of the effects of the simulated flood for Kochi. If the authors believe their own results, they should advocate to have regular floods in Kochi. • P.7, l.12. The matrix dimensions of equation 4 and 5 are inconsistent. In (4), first row, unity columns should be added behind the Z-matrices. In (4), second row, unity rows should be added in front

of the Z-matrices. Consequently, (5) is inconsistent at the matrix level. It is only correct for the overall matrix totals. ⇢ P.3, l.7 & elsewhere. Write input coefficients instead of technical coefficients, because you are dealing with an open economy in which: input coefficients = trade origin ratios * technical coefficients (see Oosterhaven, 2019, ch. 2). ⇢ P.8, l.17. As a consequence of the above misuse of the term technical, it is incorrectly stated that input coefficients change only gradually due to technological advancement. In fact, they also change much faster due to spatial substitution. Not taking this into account leads to overestimation of the indirect damages of garbage cleaning in Section 4.3. The authors might want to have a look at Oosterhaven and Többen (Spatial Econ An, 2017) for a solution to this problem.

Minor details: ⇢ P.2, l.14. These costs are only part of the total impact. So, better call them total direct impact. ⇢ P.3, l.5. Write Miller and Blair, 2009. ⇢ Figure 2. The pre-process related to garbage cleaning, not to reconstruction. ⇢ The text around Figure 2 fails to discuss how it differs from the quite comparable approaches of Hallegate and others (see also Koks et al, in Okuyama & Rose, Springer, 2019).

---

## Short Comment (SC1) · 10 Jan 2021

This paper is well-written and will be very qualified after minor changes.

Here are some suggestions authors can consider: 1. The IO model used is very good. But I suggest that the advantages of the IO model should be explained compared with other research methods. 2. Some of the phraseology in the paper is suggested to be checked and revised. 3. Relevant policy studies should be added. 4. I suggest that robust analysis should be carried out 5. Further extended research descriptions across regions, industries or products may be included if the data is allowed.

---

## Author Comment (AC1) · 28 Jan 2021

Thank you so much for your valuable comments. Please see my replies for each comment below;

1. The main question I have is whether there is any need to develop new virtual sectors. I would hesitate to make the statement "Since the garbage cleaning service and reconstruction for flood damages do not exist in the existing economy, the new process of consuming resources and producing the product has to be set up in the I-O table" (line 10). I find this hard to comprehend conceptually. Surely the cleaning and construction industry already exists prior to the disaster and you're merely employing

[Figure]

more resources to rebuild? As such, in accounting for the next IO table, these extra "production" from flood damages would be included.

REPLY: It is certainly that the garbage cleaning services already existed in the economy, but what we try to create is a virtual sector especially for flooding scenario. The activities and resources needed for this virtual sector are different in characteristic comparing with the usual garbage cleaning sector. We assume that manpower is obtained from the manpower sector which is taken from the final demand and truck use for removing garbage are taken from transportation sector. Therefore, the virtual garbage cleaning services sector's resource consumption and production is new.

2. Line 21, it would be good to add role of international organisations such as the World Bank.

REPLY: This method can be used in other cases which is related to the policy making of World Bank. It should be applicable to foresee and analyze banking policy.

3. Line 24-25: please cite the studies that have looked at the cost of removing garbage and space cleaning. What is the evidence on the cost of removing garbage in percentages – any previous studies?

REPLY: I would say that the damage of flooding mainly depends on the observe area type, whether it is city, agriculture field, or partially wasteland. We could not find some certain evidences that could indicate or compare as an index for this case. Thus, we obtained the data through the simulation by 2-d inundation model from available grid data in Japan. We selected an example case of bank break area which resulted in flooding in Kochi City. The percentage of damage is made by ourselves.

4. Lines 18-31: might be a good idea to add a diagram/figure to illustrate the difference between traditional IO analysis and hybrid IO.

REPLY: I will add a new figure to show the hybrid I-O table and add more explanation regarding these figures.

NHESSD

Interactive
comment

| Hybrid I-O Table | Truck | Manpower | Garbage Cleaning | Agricultural | Livestock | Forestry | Fishery | Mining | Food and Beverage | Textile | Lumber, Wood Product and Furniture | Pulp, Paper and Paper Product | Print Plate Making and Book Binding | Chemical Product | Petroleum and Coal | Ceramic, Stone and Clay |
|---|---|---|---|---|---|---|---|---|---|---|---|---|---|---|---|---|
| Truck (hr) | -322,588 | | 322,588 | | | | | | | | | | | | | |
| Manpower (hr) | | -57,545 | 57,545 | | | | | | | | | | | | | |
| Garbage cleaning service (yen) | | | | 138 | | | | | | | | | | | | |
| Agricultural | | | | 2,995 | 946 | 19 | 0 | 0 | 12,694 | 99 | 0 | 322 | 2 | 0 | 1 | 27 |
| Livestock | | | | 3,684 | 1,209 | 1 | 0 | 0 | 7,330 | 20 | 0 | 0 | 0 | 0 | 0 | 0 |
| Forestry | | | | 24 | 0 | 2,535 | 9 | 1 | 43 | 0 | 6,144 | 0 | 2 | 0 | 0 | 0 |
| Fishery | | | | 0 | 0 | 0 | 1,376 | 0 | 16,192 | 0 | 0 | 0 | 0 | 0 | 0 | 0 |
| Mining | | | | 0 | 0 | 6 | 0 | 191 | 5 | 0 | 0 | 678 | 183 | 126 | 0 | 4,988 |
| Food/Beverage | | | | 490 | 2,661 | 93 | 4,808 | 0 | 18,548 | 4 | 5 | 159 | 14 | 0 | 0 | 19 |
| Textile | | | | 277 | 14 | 24 | 1,088 | 81 | 100 | 2,393 | 44 | 772 | 6 | 0 | 6 | 102 |
| Lumber, Wood Product and Furniture | | | | 5 | 12 | 22 | 105 | 18 | 122 | 14 | 1,093 | 54 | 6 | 0 | 7 | 65 |
| Pulp, Paper and Paper Product | | | | 3,561 | 261 | 32 | 83 | 0 | 1,459 | 144 | 144 | 19,475 | 160 | 0 | 45 | 456 |
| Chemical Product | | | | 6,858 | 339 | 19 | 577 | 305 | 833 | 2,261 | 239 | 2,700 | 2,365 | 74 | 1,517 | 490 |
| Petroleum and Coal | | | | 2,865 | 61 | 260 | 4,706 | 469 | | 75 | 79 | 459 | 203 | 754 | 11 | 1,357 |
| Plastic and Rubber | | | | 1,604 | 65 | 146 | 814 | 112 | 1,820 | 285 | 155 | 2,303 | 203 | 2 | 1,772 | 265 |
| Ceramic, Stone and Clay | | | | 207 | 30 | 6 | 3 | 2 | 203 | 20 | 85 | 109 | 209 | 24 | 35 | 5,281 |
| Iron, Steel and Non-Ferrous | | | | 15 | 0 | 0 | 18 | 10 | 157 | 5 | 42 | 32 | 209 | 0 | 29 | 596 |
| Metal Product | | | | 96 | 6 | 12 | 102 | 182 | 755 | 43 | 122 | 77 | 113 | 0 | 14 | 262 |
| General Machinery | | | | 0 | 0 | 0 | 0 | 7 | 0 | 0 | 33 | 0 | 0 | 0 | 3 | 58 |
| Electrical Machinery | | | | 0 | 0 | 2 | 0 | 17 | 0 | 0 | 1 | 0 | 0 | 0 | 22 | 88 |
| Information/Communication Equipment | | | | 0 | 17 | 1 | 1 | 0 | 0 | 0 | 0 | 0 | 0 | 0 | 0 | 0 |
| Electronic Component | | | | 0 | 0 | 0 | 0 | 2 | 0 | 0 | 0 | 0 | 0 | 0 | 0 | 0 |
| Transport Machinery | | | | 1 | 1 | 1 | 65 | 16 | 1 | 0 | 2 | 1 | 0 | 0 | 0 | 5 |
| Precision Machinery | | | | 0 | 0 | 0 | 2,520 | 0 | 0 | 0 | 0 | 0 | 0 | 0 | 0 | 0 |
| Other Manufacturing | | | | 56 | 11 | 15 | 408 | 87 | 962 | 214 | 426 | 935 | 81 | 3 | 109 | 541 |
| Construction | | | | 1,289 | 61 | 39 | 111 | 104 | 145 | 79 | 28 | 794 | 78 | 20 | 53 | 814 |
| Public Work | | | | 0 | 0 | 0 | 0 | 0 | 0 | 0 | 0 | 0 | 0 | 0 | 0 | 0 |
| Electricity, Gas and Heat | | | | 831 | 285 | 62 | 288 | 904 | 1,383 | 291 | 212 | 5,561 | 385 | 40 | 154 | 4,677 |
| Water Supply | | | | 30 | 37 | 5 | 17 | 45 | 288 | 14 | 10 | 128 | 32 | 1 | 10 | 72 |
| Waste Treatment | | | | 0 | 14 | 0 | 0 | 8 | 64 | 1 | 10 | 12 | 14 | 0 | 0 | 115 |
| Trading | | | | 5,790 | 548 | 237 | 3,088 | 436 | 10,368 | 2,087 | 1,911 | 5,087 | 612 | 107 | 582 | 2,095 |
| Financial and Insurance | | | | 378 | 89 | 106 | 493 | 791 | 448 | 266 | 202 | 414 | 161 | 4 | 19 | 506 |
| Real Estate | | | | 73 | 109 | 9 | 46 | 79 | 237 | 59 | 41 | 124 | 55 | 4 | 26 | 176 |
| Transportation | 921 | | | 4,506 | 632 | 774 | 2,136 | 6,670 | 3,323 | 496 | 848 | 1,663 | 272 | 132 | 139 | 3,880 |
| Telecommunication | | | | 207 | 69 | 36 | 362 | 260 | 517 | 129 | 95 | 315 | 83 | 9 | 38 | 327 |
| Public Services | | | | 0 | 0 | 0 | 0 | 0 | 0 | 0 | 0 | 0 | 0 | 0 | 0 | 0 |
| Education and Research | | | | 12 | 2 | 32 | 99 | 95 | 514 | 294 | 41 | 936 | 646 | 3 | 158 | 1,578 |
| Medical and Healthcare | | | | 0 | 19 | 0 | 0 | 0 | 0 | 0 | 0 | 0 | 0 | 0 | 0 | 0 |
| Other Public Services | | | | 0 | 4 | 0 | 1,514 | 76 | 79 | 29 | 13 | 50 | 59 | 2 | 3 | 79 |
| Business Services | | | | 1,255 | 427 | 489 | 628 | 863 | 3,197 | 471 | 557 | 1,163 | 571 | 93 | 259 | 2,694 |
| Personal Services | | | | 0 | 8 | 5 | 60 | 4 | 552 | 2 | 2 | 6 | 0 | 0 | 0 | 4 |
| Office Supplies | | | | 21 | 12 | 30 | 65 | 48 | 64 | 19 | 8 | 38 | 8 | 1 | 1 | 51 |
| Unknown | | | | 1,284 | 185 | 251 | 825 | 141 | 303 | 34 | 106 | 77 | 69 | 32 | 9 | 412 |
| Manpower | | | 460 | 6,158 | 2,584 | 4,632 | 10,963 | 5,031 | 22,463 | 5,389 | 3,699 | 7,055 | 1,548 | 232 | 1,199 | 10,773 |

**Fig. 1.**

[Figure]

| Iron, Steel and Non-Ferrous | Metal Product | General Machinery | Electrical Machinery | Information Communication Equipment | Electronic Component | Transport Machinery | Precision Machinery | Other Manufacturing | Construction | Public Work | Electricity, Gas and Heat | Water Supply | Waste Treatment | Trading | Financial and Insurance | Real Estate | Transportation |
|---|---|---|---|---|---|---|---|---|---|---|---|---|---|---|---|---|---|
|  |  |  |  |  |  |  |  |  |  |  | 691 |  |  |  |  |  |  |
| 0 | 0 | 0 | 0 | 0 | 0 | 0 | 0 | 44 | 129 | 442 | 0 | 0 | 0 | 50 | 0 | 1 | 11 |
| 0 | 0 | 0 | 0 | 0 | 0 | 0 | 0 | 0 | 3 | 0 | 0 | 0 | 0 | 0 | 0 | 0 | 1 |
| 0 | 0 | 0 | 0 | 0 | 0 | 0 | 0 | 9 | 2 | 10 | 0 | 0 | 0 | 0 | 0 | 0 | 0 |
| 0 | 0 | 0 | 0 | 0 | 0 | 0 | 0 | 0 | 268 | 0 | 0 | 0 | 0 | 0 | 0 | 0 | 1 |
| 6,190 | 0 | 0 | 4 | 4 | 0 | 0 | 0 | 23 | 261 | 804 | 2,990 | 0 | 0 | 0 | 0 | 0 | 0 |
| 0 | 0 | 0 | 0 | 0 | 0 | 0 | 0 | 7 | 0 | 10 | 0 | 0 | 0 | 54 | 0 | 0 | 57 |
| 25 | 9 | 7 | 34 | 27 | 179 | 29 | 113 | 419 | 626 | 221 | 17 | 10 | 41 | 2,738 | 324 | 5 | 404 |
| 16 | 10 | 4 | 19 | 22 | 40 | 8 | 256 | 101 | 10,981 | 337 | 61 | 20 | 60 | 806 | 489 | 127 | 183 |
| 7 | 16 | 4 | 26 | 201 | 85 | 40 | 3 | 1,759 | 767 | 0 | 0 | 2 | 16 | 2,354 | 319 | 19 | 188 |
| 122 | 121 | 26 | 126 | 347 | 523 | 81 | 417 | 1,190 | 911 | 464 | 23 | 178 | 345 | 4 | 4 | 9 | 62 |
| 714 | 27 | 8 | 53 | 58 | 87 | 8 | 52 | 83 | 622 | 8,036 | 476 | 170 | 304 | 1,159 | 108 | 186 | 28,316 |
| 44 | 33 | 149 | 1,088 | 1,137 | 980 | 231 | 509 | 1,199 | 2,117 | 2,306 | 0 | 508 | 226 | 2,867 | 578 | 202 | 683 |
| 349 | 43 | 66 | 99 | 54 | 744 | 38 | 101 | 217 | 8,794 | 14,452 | 8 | 55 | 12 | 86 | 2 | 24 | 9 |
| 6,933 | 3,073 | 1,196 | 4,222 | 1,833 | 1,405 | 765 | 7,402 | 824 | 5,461 | 5,729 | 37 | 10 | 0 | 5 | 0 | 0 | 13 |
| 132 | 531 | 186 | 1,091 | 303 | 433 | 223 | 1,012 | 192 | 18,007 | 7,435 | 55 | 17 | 3 | 969 | 19 | 113 | 222 |
| 18 | 12 | 1,154 | 1,431 | 97 | 58 | 103 | 653 | 1 | 1,119 | 711 | 0 | 56 | 0 | 2 | 0 | 0 | 8 |
| 14 | 5 | 47 | 5,472 | 4 | 91 | 13 | 20 | 0 | 7 | 24 | 0 | 2 | 0 | 2 | 0 | 0 | 10 |
| 0 | 0 | 35 | 161 | 244 | 2 | 4 | 68 | 1 | 46 | 8 | 0 | 1 | 1 | 363 | 2 | 0 | 6 |
| 0 | 10 | 188 | 157 | 4,898 | 9,765 | 584 | 71 | 46 | 86 | 9 | 0 | 0 | 0 | 10 | 5 | 0 | 1 |
| 0 | 6 | 162 | 532 | 172 | 689 | 1,049 | 771 | 48 | 2,007 | 1,073 | 1 | 1 | 1 | 269 | 37 | 24 | 73 |
| 0 | 0 | 0 | 27 | 0 | 0 | 0 | 7,937 | 0 | 0 | 0 | 0 | 0 | 0 | 0 | 0 | 0 | 2,827 |
| 712 | 16 | 10 | 67 | 78 | 195 | 43 | 52 | 1,488 | 523 | 573 | 263 | 43 | 98 | 3,755 | 4,064 | 11 | 350 |
| 1,184 | 68 | 31 | 130 | 70 | 120 | 24 | 58 | 99 | 290 | 180 | 2,402 | 1,116 | 141 | 2,911 | 903 | 15,042 | 2,140 |
| 0 | 0 | 0 | 0 | 0 | 0 | 0 | 0 | 0 | 0 | 0 | 0 | 0 | 0 | 0 | 0 | 0 | 0 |
| 4,069 | 139 | 69 | 274 | 166 | 1,014 | 45 | 400 | 353 | 961 | 674 | 15,127 | 496 | 904 | 7,808 | 613 | 1,169 | 950 |
| 51 | 6 | 4 | 20 | 12 | 48 | 4 | 23 | 27 | 153 | 96 | 67 | 2,030 | 229 | 592 | 299 | 112 | 459 |
| 0 | 0 | 1 | 0 | 2 | 18 | 0 | 0 | 18 | 4 | 44 | 639 | 63 | 30 | 534 | 429 | 2 | 321 |
| 1,222 | 712 | 444 | 1,764 | 1,247 | 1,590 | 442 | 2,374 | 2,688 | 16,696 | 11,821 | 227 | 289 | 334 | 8,656 | 1,204 | 507 | 6,504 |
| 218 | 109 | 59 | 210 | 242 | 212 | 30 | 306 | 364 | 1,926 | 3,666 | 1,610 | 67 | 174 | 6,577 | 8,957 | 23,441 | 4,206 |
| 78 | 55 | 34 | 77 | 25 | 53 | 19 | 63 | 74 | 1,271 | 331 | 720 | 34 | 55 | 11,141 | 2,184 | 8,240 | 5,295 |
| 1,207 | 337 | 193 | 663 | 416 | 628 | 142 | 654 | 3,421 | 7,919 | 8,688 | 784 | 226 | 1,265 | 21,132 | 5,539 | 760 | 24,303 |
| 129 | 73 | 85 | 319 | 164 | 294 | 75 | 121 | 192 | 1,375 | 1,820 | 902 | 547 | 204 | 14,490 | 8,366 | 1,117 | 2,271 |
| 0 | 0 | 0 | 0 | 0 | 0 | 0 | 0 | 0 | 0 | 0 | 0 | 0 | 0 | 0 | 0 | 0 | 0 |
| 1,125 | 94 | 223 | 1,444 | 1,758 | 3,022 | 439 | 596 | 1,008 | 253 | 503 | 753 | 4 | 5 | 1,935 | 144 | 0 | 300 |
| 0 | 0 | 0 | 0 | 0 | 0 | 0 | 0 | 0 | 0 | 0 | 0 | 5 | 0 | 9 | 24 | 2 | 55 |
| 20 | 10 | 12 | 39 | 17 | 26 | 2 | 22 | 29 | 180 | 514 | 408 | 791 | 45 | 784 | 1,703 | 370 | 216 |
| 2,084 | 248 | 538 | 1,251 | 522 | 1,592 | 265 | 715 | 1,204 | 15,891 | 27,655 | 9,626 | 1,653 | 1,133 | 21,194 | 14,341 | 5,152 | 29,471 |
| 4 | 0 | 0 | 1 | 2 | 7 | 1 | 2 | 3 | 38 | 77 | 8 | 4 | 1 | 599 | 40 | 218 | 74 |
| 9 | 3 | 11 | 28 | 32 | 19 | 7 | 17 | 52 | 79 | 407 | 7 | 10 | 70 | 930 | 681 | 91 | 353 |
| 179 | 24 | 112 | 299 | 41 | 31 | 21 | 208 | 50 | 2,632 | 2,242 | 150 | 136 | 26 | 3,057 | 878 | 1,780 | 1,148 |
| 3,792 | 2,429 | 2,518 | 8,202 | 3,240 | 3,106 | 1,024 | 4,999 | 6,976 | 63,578 | 68,102 | 12,212 | 2,162 | 10,240 | 181,109 | 61,074 | 18,298 | 53,516 |

**Fig. 2.**

| Telecom munication | Public Services | Education and Research | Medical and Healthcare | Other Public Services | Business Services | Personal Services | Office Supplies | Unknown | Manpower | Final demand | Total Production |
|---|---|---|---|---|---|---|---|---|---|---|---|
| | | | | | | | | | | 0 | 0 |
| | | | | | | | | | | 0 | 0 |
| | | | | | | | | | | 552 | 1,381 |
| 0 | 4 | 9 | 767 | 47 | 2 | 2,799 | 0 | 0 | -17,004 | 82,252 | 86,658 |
| 0 | 0 | 78 | 116 | 0 | 0 | 769 | 0 | 0 | 5,077 | -4,678 | 13,610 |
| 0 | 1 | 0 | 30 | 0 | 0 | 150 | 0 | 0 | 3,691 | 6,568 | 19,219 |
| 0 | 1 | 0 | 225 | 0 | 0 | 1,323 | 0 | 0 | -7,029 | 39,589 | 51,946 |
| 0 | 1 | 3 | 0 | 0 | 0 | -3 | 0 | 2 | 4,291 | 1,053 | 21,800 |
| 0 | 74 | 108 | 3,527 | 36 | 1 | 26,058 | 0 | 29 | -26,413 | 92,743 | 123,092 |
| 104 | 1,621 | 60 | 1,638 | 1,676 | 330 | 897 | 105 | 17 | 6,735 | -4,944 | 18,374 |
| 286 | 343 | 270 | 1,517 | 590 | 214 | 1,078 | 0 | 4 | 6,667 | -7,030 | 19,002 |
| 2,145 | 95 | 522 | 1,266 | 147 | 455 | 723 | 2,735 | 48 | -4,682 | 23,887 | 58,967 |
| 256 | 262 | 378 | 58,365 | 86 | 620 | 1,299 | 102 | 307 | 78,404 | -152,789 | 10,820 |
| 154 | 2,513 | 753 | 1,768 | 219 | 508 | 1,997 | 0 | 782 | 59,639 | -118,561 | 2,148 |
| 240 | 447 | 297 | 869 | 251 | 2,033 | 603 | 478 | 193 | 24,792 | -47,554 | 7,052 |
| 1 | 59 | 295 | 308 | 17 | 263 | 284 | 35 | 185 | 734 | 22,000 | 55,548 |
| 14 | 40 | 4 | 610 | 8 | 148 | 98 | 6 | 461 | 14,555 | -12,035 | 43,930 |
| 52 | 972 | 22 | 143 | 92 | 218 | 579 | 2 | 100 | 29,117 | -54,667 | 9,357 |
| 1 | 57 | 0 | 0 | 0 | 1,018 | 2 | 0 | 0 | 1,544 | 273 | 8,419 |
| 0 | 3 | 0 | 0 | 0 | 1,677 | 2 | 0 | 0 | -13,605 | 40,453 | 34,371 |
| 17 | 1,685 | 0 | 4,435 | 0 | 901 | 1,753 | 165 | 0 | -4,278 | 13,396 | 19,035 |
| 107 | 362 | 66 | 2 | 0 | 1,957 | 2 | 187 | 0 | -5,435 | 16,818 | 29,898 |
| 62 | 474 | 128 | 76 | 4 | 1,426 | 87 | 0 | 33 | 4,563 | -7,163 | 6,698 |
| 0 | 1,760 | 10 | 0 | 0 | 12,259 | 4 | 0 | 0 | 2,330 | 3,336 | 33,010 |
| 2,994 | 3,283 | 2,446 | 2,056 | 1,691 | 1,528 | 1,780 | 897 | 40 | 15,414 | -20,053 | 28,265 |
| 1,088 | 6,746 | 2,026 | 2,364 | 172 | 405 | 1,300 | 0 | 0 | -57,549 | 194,142 | 181,218 |
| 0 | 0 | 0 | 0 | 0 | 0 | 0 | 0 | 0 | -101,957 | 290,009 | 188,052 |
| 1,029 | 1,627 | 3,697 | 4,805 | 127 | 963 | 4,123 | 0 | 232 | 30,122 | -26,105 | 70,924 |
| 379 | 776 | 1,470 | 2,612 | 83 | 122 | 2,342 | 0 | 78 | 4,272 | -402 | 16,653 |
| 495 | 7,403 | 749 | 1,205 | 1 | 43 | 3,707 | 0 | 94 | 10,347 | -5,857 | 20,530 |
| 1,803 | 3,274 | 2,019 | 24,961 | 1,584 | 4,336 | 22,630 | 1,486 | 364 | 36,284 | 210,888 | 401,298 |
| 748 | 10,750 | 170 | 2,586 | 1,738 | 1,690 | 1,797 | 0 | 115 | 23,589 | 73,002 | 172,436 |
| 2,249 | 312 | 663 | 8,570 | 749 | 1,186 | 3,276 | 0 | 933 | -9,999 | 276,599 | 315,325 |
| 3,281 | 7,435 | 3,916 | 6,611 | 1,188 | 2,504 | 8,372 | 313 | 1,959 | 27,836 | 24,590 | 191,753 |
| 24,514 | 6,759 | 2,762 | 6,308 | 2,508 | 8,560 | 5,779 | 0 | 1,013 | 30,457 | 6,420 | 130,071 |
| 0 | 0 | 0 | 0 | 0 | 0 | 0 | 0 | 5,247 | -67,994 | 323,311 | 260,564 |
| 2,295 | 34 | 426 | 1,881 | 0 | 407 | 144 | 0 | 922 | -9,615 | 161,242 | 175,752 |
| 82 | 5 | 0 | 12,426 | 0 | 2 | 11 | 0 | 77 | -167,494 | 600,057 | 445,280 |
| 227 | 1 | 121 | 1,347 | 0 | 356 | 2,920 | 0 | 48 | -4,150 | 25,706 | 33,672 |
| 15,286 | 12,932 | 7,192 | 14,097 | 2,775 | 14,596 | 7,919 | 0 | 1,262 | 160,719 | -232,478 | 151,502 |
| 1,670 | 168 | 169 | 10,103 | 116 | 230 | 4,018 | 0 | 72 | -93,298 | 330,294 | 255,264 |
| 264 | 813 | 445 | 1,009 | 169 | 211 | 427 | 0 | 4 | 0 | 0 | 6,514 |
| 894 | 149 | 2,466 | 1,608 | 192 | 1,373 | 517 | 3 | 0 | 9,323 | -9,329 | 23,938 |
| 26,095 | 94,253 | 120,285 | 215,915 | 16,091 | 53,686 | 82,063 | 0 | 882 | 0 | 0 | 1,197,573 |

**Fig. 3.**

---

## Author Comment (AC2) · 4 Feb 2021

Thank you for your comments. I understand that you may have some question regarding how to construct the hybrid I-O analysis so I will certainly add more data to show how the table look like. Please my replies below;

1. The IO model used is very good. But I suggest that the advantages of the IO model should be explained compared with other research methods.

REPLY: The hybrid I-O model is unique comparing to the conventional I-O model. It could analyze both monetary value and physical value in one table. The hybrid of
physical and monetary, in other words, the integration of four submatrices (P, M, Cu, and Cd) is an illustration of the connection between a new industry and the existing economy, where the transferred resources are indicated along the way. Moreover, with the principal of physical and monetary, the resource price alteration could be made separately from the existing economy's price, which is the further approach to pricing policy.

2. Some of the phraseology in the paper is suggested to be checked and revised.

REPLY: Noted. I will make a revision.

3. Relevant policy studies should be added.

REPLY: The hybrid I-O analysis method leads to the change of economic structure. It is accounted for both early phase change for garbage cleaning activities and thereafter the reconstruction and maintenance phase. If we could predict the short-term change of the economic structure, we could adapt for a new policy in order to remediate the effect of this disaster.

4. I suggest that robust analysis should be carried out

REPLY: I will add more instruction how to construct the hybrid I-O analysis.

5. Further extended research descriptions across regions, industries or products may be included if the data is allowed.

REPLY: I will add the hybrid I-O table to the supplementary.

NHESSD
| Hybrid I-O Table                   | Truck       | Manpower | Garbage
Cleaning | Agricultural | Livestock | Forestry | Fishery | Mining | Food and
Beverage | Textile | Lumber, Wood
Product
and Furniture | Pulp, Paper and
Paper Product | Print Plate
Making and
Book Binding | Chemical
Product | Petroleum and
Coal | Ceramic, Stone
and Clay |
|------------------------------------|-------------|----------|---------------------|--------------|-----------|----------|---------|--------|----------------------|---------|------------------------------------------|----------------------------------|-------------------------------------------|---------------------|-----------------------|----------------------------|
| Truck (hr)                         | -322.588    |          | 322,588             |              |           |          |         |        |                      |         |                                          |                                  |                                           |                     |                       |                            |
| Manpower (hr)                      |             | -57,545  | 57,545              |              |           |          |         |        |                      |         |                                          |                                  |                                           |                     |                       |                            |
| Garbage cleaning service (yes      | n)          |          |                     | 138          |           |          |         |        |                      |         |                                          |                                  |                                           |                     |                       |                            |
| Agricultural                       |             |          |                     | 2,995        | 946       | 19       | 0       | 0      | 12,694               | 99      | 0                                        | 322                              | 2                                         | 0                   | 1                     | 27                         |
| Livestock                          |             |          |                     | 3,684        | 1,209     | 1        | 0       | 0      | 7,330                | 20      | 0                                        | 0                                | 0                                         | 0                   | 0                     | 0                          |
| Forestry                           |             |          |                     | 24           | 0         | 2,535    | 9       | 1      | 43                   | 0       | 6,144                                    | 0                                | 2                                         | 0                   | 0                     | 0                          |
| Fishery                            |             |          |                     | 0            | 0         | 0        | 1,376   | 0      | 16,192               | 0       | 0                                        | 0                                | 0                                         | 0                   | 0                     | 0                          |
| Mining                             |             |          |                     | 0            | 0         | 6        | 0       | 191    | 5                    | 0       | 0                                        | 678                              | 183                                       | 126                 | 0                     | 4,988                      |
| Food/Beverage                      |             |          |                     | 490          | 2,661     | 93       | 4,808   | 0      | 18,548               | 4       | 5                                        | 159                              | 14                                        | 0                   | 0                     | 19                         |
| Textile                            |             |          |                     | 277          | 14        | 24       | 1,088   | 81     | 100                  | 2,393   | 44                                       | 772                              | 6                                         | 0                   | 6                     | 102                        |
| Lumber, Wood Product and Furniture |             |          |                     | 5            | 12        | 22       | 105     | 18     | 122                  | 14      | 1,093                                    | 54                               | 6                                         | 0                   | 7                     | 65                         |
| Pulp, Paper and Paper Prod         | uct         |          |                     | 3,561        | 261       | 32       | 83      | 0      | 1,459                | 144     | 144                                      | 19,475                           | 160                                       | 0                   | 45                    | 456                        |
| Chemical Product                   |             |          |                     | 6,858        | 339       | 19       | 577     | 305    | 833                  | 2,261   | 239                                      | 2,700                            | 2,365                                     | 74                  | 1,517                 | 490                        |
| Petroleum and Coal                 |             |          |                     | 2,865        | 61        | 260      | 4,706   | 610    | 469                  | 75      | 79                                       | 459                              | 203                                       | 754                 | 11                    | 1,357                      |
| Plastic and Rubber                 |             |          |                     | 1,604        | 65        | 146      | 814     | 112    | 1,820                | 285     | 155                                      | 2,303                            | 203                                       | 2                   | 1,772                 | 265                        |
| Ceramic, Stone and Clay            |             |          |                     | 207          | 30        | 6        | 3       | 2      | 203                  | 20      | 85                                       | 109                              | 209                                       | 24                  | 35                    | 5,281                      |
| Iron, Steel and Non-Ferrous        |             |          |                     | 15           | 0         | 0        | 18      | 10     | 157                  | 5       | 42                                       | 32                               | 209                                       | 0                   | 29                    | 596                        |
| Metal Product                      |             |          |                     | 96           | 6         | 12       | 102     | 182    | 755                  | 43      | 122                                      | 77                               | 113                                       | 0                   | 14                    | 262                        |
| General Machinery                  |             |          |                     | 0            | 0         | 0        | 0       | 7      | 0                    | 0       | 33                                       | 0                                | 0                                         | 0                   | 3                     | 58                         |
| Electrical Machinery               |             |          |                     | 0            | 0         | 2        | 0       | 17     | 0                    | 0       | 1                                        | 0                                | 0                                         | 0                   | 22                    | 88                         |
| Information/Communicatio           | n Equipment |          |                     | 0            | 17        | 1        | 1       | 0      | 0                    | 0       | 0                                        | 0                                | 0                                         | 0                   | 0                     | 0                          |
| Electronic Component               |             |          |                     | 0            | 0         | 0        | 0       | 2      | 0                    | 0       | 0                                        | 0                                | 0                                         | 0                   | 0                     | 0                          |
| Transport Machinery                |             |          |                     | 1            | 1         | 1        | 65      | 16     | 1                    | 0       | 2                                        | 1                                | 0                                         | 0                   | 0                     | 5                          |
| Precision Machinery                |             |          |                     | 0            | 0         | 0        | 2,520   | 0      | 0                    | 0       | 0                                        | 0                                | 0                                         | 0                   | 0                     | 0                          |
| Other Manufacturing                |             |          |                     | 56           | 11        | 15       | 408     | 87     | 962                  | 214     | 426                                      | 935                              | 81                                        | 3                   | 109                   | 541                        |
| Construction                       |             |          |                     | 1,289        | 61        | 39       | 111     | 104    | 145                  | 79      | 28                                       | 794                              | 78                                        | 20                  | 53                    | 814                        |
| Public Work                        |             |          |                     | 0            | 0         | 0        | 0       | 0      | 0                    | 0       | 0                                        | 0                                | 0                                         | 0                   | 0                     | 0                          |
| Electricity, Gas and Heat          |             |          |                     | 831          | 285       | 62       | 288     | 904    | 1,383                | 291     | 212                                      | 5,561                            | 385                                       | 40                  | 154                   | 4,677                      |
| Water Supply                       |             |          |                     | 30           | 37        | 5        | 17      | 45     | 288                  | 14      | 10                                       | 128                              | 32                                        | 1                   | 10                    | 72                         |
| Waste Treatment                    |             |          |                     | 0            | 14        | 0        | 0       | 8      | 64                   | 1       | 10                                       | 12                               | 14                                        | 0                   | 0                     | 115                        |
| Trading                            |             |          |                     | 5,790        | 548       | 237      | 3,088   | 436    | 10,368               | 2,087   | 1,911                                    | 5,087                            | 612                                       | 107                 | 582                   | 2,095                      |
| Financial and Insurance            |             |          |                     | 378          | 89        | 106      | 493     | 791    | 448                  | 266     | 202                                      | 414                              | 161                                       | 4                   | 19                    | 506                        |
| Real Estate                        |             |          |                     | 73           | 109       | 9        | 46      | 79     | 237                  | 59      | 41                                       | 124                              | 55                                        | 4                   | 26                    | 176                        |
| Transportation                     | 921         |          |                     | 4,506        | 632       | 774      | 2,136   | 6,670  | 3,323                | 496     | 848                                      | 1,663                            | 272                                       | 132                 | 139                   | 3,880                      |
| Telecommunication                  |             |          |                     | 207          | 69        | 36       | 362     | 260    | 517                  | 129     | 95                                       | 315                              | 83                                        | 9                   | 38                    | 327                        |
| Public Services                    |             |          |                     | 0            | 0         | 0        | 0       | 0      | 0                    | 0       | 0                                        | 0                                | 0                                         | 0                   | 0                     | 0                          |
| Education and Research             |             |          |                     | 12           | 2         | 32       | 99      | 95     | 514                  | 294     | 41                                       | 936                              | 646                                       | 3                   | 158                   | 1,578                      |
| Medical and Healthcare             |             |          |                     | 0            | 19        | 0        | 0       | 0      | 0                    | 0       | 0                                        | 0                                | 0                                         | 0                   | 0                     | 0                          |
| Other Public Services              |             |          |                     | 0            | 4         | 0        | 1,514   | 76     | 79                   | 29      | 13                                       | 50                               | 59                                        | 2                   | 3                     | 79                         |
| Business Services                  |             |          |                     | 1,255        | 427       | 489      | 628     | 863    | 3,197                | 471     | 557                                      | 1,163                            | 571                                       | 93                  | 259                   | 2,694                      |
| Personal Services                  |             |          |                     | 0            | 8         | 5        | 60      | 4      | 552                  | 2       | 2                                        | 6                                | 0                                         | 0                   | 0                     | 4                          |
| Office Supplies                    |             |          |                     | 21           | 12        | 30       | 65      | 48     | 64                   | 19      | 8                                        | 38                               | 8                                         | 1                   | 1                     | 51                         |
| Unknown                            |             |          |                     | 1,284        | 185       | 251      | 825     | 141    | 303                  | 34      | 106                                      | 77                               | 69                                        | 32                  | 9                     | 412                        |
| Manpower                           |             | 460      |                     | 6,158        | 2,584     | 4,632    | 10,963  | 5,031  | 22,463               | 5,389   | 3,699                                    | 7,055                            | 1,548                                     | 232                 | 1,199                 | 10,773                     |

Fig. 1.
| Iron, Steel and
Non-Ferrous | Metal
Product | General
Machinery | Electrical
Machinery | Information
Communication
Equipment | Electronic
Component | Transport
Machinery | Precision
Machinery | Other
Manufacturing | Construction | Public Work | Electricity, Gas
and Heat | Water Supply | Waste
Treatment | Trading | Financial
and Insurance | Real Estate | Transportation |
|--------------------------------|------------------|----------------------|-------------------------|-------------------------------------------|-------------------------|------------------------|------------------------|------------------------|--------------|-------------|------------------------------|--------------|--------------------|---------|----------------------------|-------------|----------------|
|                                |                  |                      |                         |                                           |                         |                        |                        |                        |              | 691         |                              |              |                    |         |                            |             |                |
| 0                              | 0                | 0                    | 0                       | 0                                         | 0                       | 0                      | 0                      | 44                     | 129          | 442         | 0                            | 0            | 0                  | 50      | 0                          | 1           | 11             |
| 0                              | 0                | 0                    | 0                       | 0                                         | 0                       | 0                      | 0                      | 3                      | 0            | 0           | 0                            | 0            | 0                  | 0       | 0                          | 0           | 1              |
| 0                              | 0                | 0                    | 0                       | 0                                         | 0                       | 0                      | 0                      | 9                      | 2            | 10          | 0                            | 0            | 0                  | 0       | 0                          | 0           | 0              |
| 0                              | 0                | 0                    | 0                       | 0                                         | 0                       | 0                      | 0                      | 268                    | 0            | 0           | 0                            | 0            | 0                  | 0       | 0                          | 0           | 1              |
| 6,190                          | 0                | 0                    | 4                       | 4                                         | 0                       | 0                      | 0                      | 23                     | 261          | 804         | 2,990                        | 0            | 0                  | 0       | 0                          | 0           | 0              |
| 0                              | 0                | 0                    | 0                       | 0                                         | 0                       | 0                      | 0                      | 7                      | 0            | 10          | 0                            | 0            | 0                  | 54      | 0                          | 0           | 57             |
| 25                             | 9                | 7                    | 34                      | 27                                        | 179                     | 29                     | 113                    | 419                    | 626          | 221         | 17                           | 10           | 41                 | 2,738   | 324                        | 5           | 404            |
| 16                             | 10               | 4                    | 19                      | 22                                        | 40                      | 8                      | 256                    | 101                    | 10,981       | 337         | 61                           | 20           | 60                 | 806     | 489                        | 127         | 183            |
| 7                              | 16               | 4                    | 26                      | 201                                       | 85                      | 40                     | 3                      | 1,759                  | 767          | 0           | 0                            | 2            | 16                 | 2,354   | 319                        | 19          | 188            |
| 122                            | 121              | 26                   | 126                     | 347                                       | 523                     | 81                     | 417                    | 1,190                  | 911          | 464         | 23                           | 178          | 345                | 4       | 4                          | 9           | 62             |
| 714                            | 27               | 8                    | 53                      | 58                                        | 87                      | 8                      | 52                     | 83                     | 622          | 8,036       | 476                          | 170          | 304                | 1,159   | 108                        | 186         | 28,316         |
| 44                             | 33               | 149                  | 1,088                   | 1,137                                     | 980                     | 231                    | 509                    | 1,199                  | 2,117        | 2,306       | 0                            | 508          | 226                | 2,867   | 578                        | 202         | 683            |
| 349                            | 43               | 66                   | 99                      | 54                                        | 744                     | 38                     | 101                    | 217                    | 8,794        | 14,452      | 8                            | 55           | 12                 | 86      | 2                          | 24          | 9              |
| 6,933                          | 3,073            | 1,196                | 4,222                   | 1,833                                     | 1,405                   | 765                    | 7,402                  | 824                    | 5,461        | 5,729       | 37                           | 10           | 0                  | 5       | 0                          | 0           | 13             |
| 132                            | 531              | 186                  | 1,091                   | 303                                       | 433                     | 223                    | 1,012                  | 192                    | 18,007       | 7,435       | 55                           | 17           | 3                  | 969     | 19                         | 113         | 222            |
| 18                             | 12               | 1,154                | 1,431                   | 97                                        | 58                      | 103                    | 653                    | 1                      | 1,119        | 711         | 0                            | 56           | 0                  | 2       | 0                          | 0           | 8              |
| 14                             | 5                | 47                   | 5,472                   | 4                                         | 91                      | 13                     | 20                     | 0                      | 7            | 24          | 0                            | 2            | 0                  | 2       | 0                          | 0           | 10             |
| 0                              | 0                | 35                   | 161                     | 244                                       | 2                       | 4                      | 68                     | 1                      | 46           | 8           | 0                            | 1            | 1                  | 363     | 2                          | 0           | 6              |
| 0                              | 10               | 188                  | 157                     | 4,898                                     | 9,765                   | 584                    | 71                     | 46                     | 86           | 9           | 0                            | 0            | 0                  | 10      | 5                          | 0           | 1              |
| 0                              | 6                | 162                  | 532                     | 172                                       | 689                     | 1,049                  | 771                    | 48                     | 2,007        | 1,073       | 1                            | 1            | 1                  | 269     | 37                         | 24          | 73             |
| 0                              | 0                | 0                    | 27                      | 0                                         | 0                       | 0                      | 7,937                  | 0                      | 0            | 0           | 0                            | 0            | 0                  | 0       | 0                          | 0           | 2,827          |
| 712                            | 16               | 10                   | 67                      | 78                                        | 195                     | 43                     | 52                     | 1,488                  | 523          | 573         | 263                          | 43           | 98                 | 3,755   | 4,064                      | 11          | 350            |
| 1,184                          | 68               | 31                   | 130                     | 70                                        | 120                     | 24                     | 58                     | 99                     | 290          | 180         | 2,402                        | 1,116        | 141                | 2,911   | 903                        | 15,042      | 2,140          |
| 0                              | 0                | 0                    | 0                       | 0                                         | 0                       | 0                      | 0                      | 0                      | 0            | 0           | 0                            | 0            | 0                  | 0       | 0                          | 0           | 0              |
| 4,069                          | 139              | 69                   | 274                     | 166                                       | 1,014                   | 45                     | 400                    | 353                    | 961          | 674         | 15,127                       | 496          | 904                | 7,808   | 613                        | 1,169       | 950            |
| 51                             | 6                | 4                    | 20                      | 12                                        | 48                      | 4                      | 23                     | 27                     | 153          | 96          | 67                           | 2,030        | 229                | 592     | 299                        | 112         | 459            |
| 0                              | 0                | 1                    | 0                       | 2                                         | 18                      | 0                      | 18                     | 4                      | 44           | 639         | 63                           | 30           | 0                  | 534     | 429                        | 2           | 321            |
| 1,222                          | 712              | 444                  | 1,764                   | 1,247                                     | 1,590                   | 442                    | 2,374                  | 2,688                  | 16,696       | 11,821      | 227                          | 289          | 334                | 8,656   | 1,204                      | 507         | 6,504          |
| 218                            | 109              | 59                   | 210                     | 242                                       | 212                     | 30                     | 306                    | 364                    | 1,926        | 3,666       | 1,610                        | 67           | 174                | 6,577   | 8,957                      | 23,441      | 4,206          |
| 78                             | 55               | 34                   | 77                      | 25                                        | 53                      | 19                     | 63                     | 74                     | 1,271        | 331         | 720                          | 34           | 55                 | 11,141  | 2,184                      | 8,240       | 5,295          |
| 1,207                          | 337              | 193                  | 663                     | 416                                       | 628                     | 142                    | 654                    | 3,421                  | 7,919        | 8,688       | /84                          | 226          | 1,265              | 21,132  | 5,539                      | 760         | 24,303         |
| 129                            | 73               | 85                   | 319                     | 164                                       | 294                     | 75                     | 121                    | 192                    | 1,375        | 1,820       | 902                          | 547          | 204                | 14,490  | 8,366                      | 1,117       | 2,271          |
| 0                              | 0                | 0                    | 0                       | 0                                         | 0                       | 0                      | 0                      | 0                      | 0            | 0           | 0                            | 0            | 0                  | 0       | 0                          | 0           | 0              |
| 1,125                          | 94               | 223                  | 1,444                   | 1,758                                     | 3,022                   | 439                    | 596                    | 1,008                  | 253          | 503         | 753                          | 4            | 5                  | 1,935   | 144                        | 0           | 300            |
| 0                              | 0                | 0                    | 0                       | 0                                         | 0                       | 0                      | 0                      | 0                      | 0            | 0           | 0                            | 5            | 0                  | 9       | 24                         | 2           | 55             |
| 20                             | 10               | 12                   | 39                      | 17                                        | 26                      | 2                      | 22                     | 29                     | 180          | 514         | 408                          | 791          | 45                 | 784     | 1,703                      | 370         | 216            |
| 2,084                          | 248              | 538                  | 1,251                   | 522                                       | 1,592                   | 265                    | /15                    | 1,204                  | 15,891       | 27,655      | 9,626                        | 1,653        | 1,133              | 21,194  | 14,341                     | 5,152       | 29,4/1         |
| 4                              | 0                | 0                    | 1                       | 2                                         | /                       | 1                      | 2                      | 3                      | 38           | 11          | 8                            | 4            | 1                  | 599     | 40                         | 218         | /4             |
| 9                              | 3                | 11                   | 28                      | 32                                        | 19                      | 7                      | 1/                     | 52                     | 79           | 407         | 1                            | 10           | 70                 | 930     | 681                        | 91          | 353            |
| 1/9                            | 24               | 112                  | 299                     | 41                                        | 31                      | 21                     | 208                    | 00                     | 2,032        | 2,242       | 150                          | 150          | 20                 | 5,057   | 6/6                        | 1,780       | 1,148          |
| 5,792                          | 2,429            | 2,518                | 6,202                   | 5,240                                     | 5,100                   | 1,024                  | 4,999                  | 6,976                  | 03,578       | 66,102      | 12,212                       | 2,102        | 10,240             | 181,109 | 61,074                     | 16,298      | 55,510         |

**Fig. 2.**

| NILSSD |
|--------|
|--------|
| Telecom
munication | Public
Services | Education
and Research | Medical
and Healthcare | Other Public
Services | Business
Services | Personal
Services | Office
Supplies | Unknown | Manpower | Final demand | Total
Production |
|-----------------------|--------------------|---------------------------|---------------------------|--------------------------|----------------------|----------------------|--------------------|---------|----------|--------------|---------------------|
|                       |                    |                           |                           |                          |                      |                      |                    |         |          | 0            | 0                   |
|                       |                    |                           |                           |                          |                      |                      |                    |         |          | 0            | 0                   |
| 0                     |                    | 0                         | 767                       | 47                       | 2                    | 2 700                | 0                  | 0       | 17.004   | 552          | 1,381               |
| 0                     | 4                  | 9                         | /6/                       | 47                       | 2                    | 2,799                | 0                  | 0       | -17,004  | 82,252       | 86,658              |
| 0                     | 0                  | /8                        | 110                       | 0                        | 0                    | 769                  | 0                  | 0       | 3,077    | -4,070       | 13,610              |
| 0                     | 1                  | 0                         | 30                        | 0                        | 0                    | 1 2 2 2              | 0                  | 0       | 3,031    | 20,508       | 19,219              |
| 0                     | 1                  | 2                         | 223                       | 0                        | 0                    | 1,525                | 0                  | 0       | -7,029   | 39,369       | 51,946              |
| 0                     | 74                 | 109                       | 2 5 2 7                   | 36                       | 1                    | 26.059               | 0                  | 20      | 4,231    | 1,033        | 122,800             |
| 104                   | 1 621              | 108                       | 1,527                     | 1 676                    | 220                  | 20,038               | 105                | 17      | -20,413  | 92,743       | 123,092             |
| 104                   | 242                | 270                       | 1,038                     | 1,070                    | 330                  | 1.079                | 105                | 1/      | 6,733    | 7,020        | 10,574              |
| 2 1 4 5               | 95                 | £270                      | 1,517                     | 147                      | 455                  | 722                  | 2 725              | 4       | 4 692    | 22 997       | 19,002              |
| 2,143                 | 262                | 379                       | 58 365                    | 147                      | 433                  | 1 200                | 2,733              | 48      | 78 404   | -152 789     | 10,907              |
| 154                   | 202                | 752                       | 1 769                     | 219                      | 620                  | 1,299                | 102                | 792     | F0 620   | -132,785     | 10,820              |
| 240                   | 2,515              | 755                       | 1,708                     | 215                      | 2 022                | 1,557                | 479                | 102     | 33,033   | 47 554       | 2,140               |
| 240                   | 447                | 297                       | 309                       | 17                       | 2,033                | 284                  | 478                | 195     | 24,732   | 22,000       | 7,052               |
| 14                    | 40                 | 255                       | 610                       |                          | 1/18                 | 284                  | 55                 | 461     | 14 555   | -12 035      | 12 020              |
| 52                    | 972                |                           | 143                       | 92                       | 219                  | 570                  | 2                  | 100     | 29 117   | -12,055      | 43,930              |
| 32                    | 572                | 22                        | 145                       | 52                       | 1 019                | 375                  | 2                  | 100     | 1 544    | -54,007      | 9,337               |
| 1                     | 37                 | 0                         | 0                         | 0                        | 1,018                | 2                    | 0                  | 0       | -13 605  | 40.453       | 0,419               |
| 17                    | 1 695              | 0                         | 4 425                     | 0                        | 1,077                | 1 752                | 165                | 0       | -13,603  | 40,433       | 54,571              |
| 107                   | 1,085              | 66                        | 4,433                     | 0                        | 1 957                | 1,733                | 103                | 0       | -4,278   | 16,919       | 19,033              |
| 62                    | 302                | 128                       | 76                        | 0                        | 1,937                | 2                    | 187                | 22      | -3,433   | 7 162        | 29,898              |
| 02                    | 1 760              | 128                       | /0                        | 4                        | 12 250               | 87                   | 0                  |         | 4,505    | -7,103       | 0,058               |
| 2 994                 | 2,700              | 2 446                     | 2 056                     | 1 601                    | 1 5 2 9              | 1 790                | 907                | 40      | 15 414   | 20.052       | 33,010              |
| 1.088                 | 5,285              | 2,440                     | 2,050                     | 1,051                    | 1,528                | 1,780                | 0                  | 40      | -57 549  | 194 142      | 191 219             |
| 1,000                 | 0,740              | 2,020                     | 2,504                     | 1/2                      | 0                    | 1,500                | 0                  | 0       | 101.957  | 290,009      | 101,210             |
| 1 0 2 9               | 1 627              | 3 697                     | 4 805                     | 127                      | 963                  | 4 123                | 0                  | 232     | 30 122   | -26 105      | 70,924              |
| 379                   | 776                | 1 470                     | 2 612                     | 83                       | 122                  | 2 342                | 0                  | 78      | 4 272    | -20,103      | 16 653              |
| /95                   | 7 403              | 7/9                       | 1 205                     | 1                        | 13                   | 3 707                | 0                  | 94      | 10 347   | -5 857       | 20,530              |
| 1 803                 | 3 274              | 2 019                     | 24 961                    | 1 584                    | 4 336                | 22,630               | 1 486              | 364     | 36 284   | 210 888      | 401 298             |
| 748                   | 10 750             | 170                       | 2 586                     | 1,304                    | 1,690                | 1 797                | 1,400              | 115     | 23 589   | 73 002       | 172 436             |
| 2 249                 | 312                | 663                       | 8 570                     | 749                      | 1,050                | 3 276                | 0                  | 933     | -9 999   | 276 599      | 315 325             |
| 3 281                 | 7 435              | 3 916                     | 6 611                     | 1 188                    | 2 504                | 8 372                | 313                | 1 959   | 27.836   | 24 590       | 191 753             |
| 24.514                | 6,759              | 2,762                     | 6.308                     | 2,508                    | 8,560                | 5,779                | 0                  | 1,013   | 30,457   | 6.420        | 130 071             |
| 0                     | 0                  | 0                         | 0                         | 0                        | 0                    | 0                    | 0                  | 5,247   | -67,994  | 323,311      | 260 564             |
| 2,295                 | 34                 | 426                       | 1.881                     | 0                        | 407                  | 144                  | 0                  | 977     | -9,615   | 161,242      | 175,752             |
| 82                    | 5                  | 0                         | 12,426                    | 0                        | 2                    | 11                   | 0                  | 77      | -167,494 | 600.057      | 445,280             |
| 227                   | 1                  | 121                       | 1,347                     | 0                        | 356                  | 2,920                | 0                  | 48      | -4,150   | 25,706       | 33,672              |
| 15,286                | 12,932             | 7,192                     | 14,097                    | 2,775                    | 14,596               | 7,919                | 0                  | 1,262   | 160,719  | -232,478     | 151,502             |
| 1.670                 | 168                | 169                       | 10,103                    | 116                      | 230                  | 4.018                | 0                  | 72      | -93,298  | 330,294      | 255.264             |
| 264                   | 813                | 445                       | 1,009                     | 169                      | 211                  | 427                  | 0                  | 4       | 0        | 0            | 6,514               |
| 894                   | 149                | 2,466                     | 1,608                     | 192                      | 1,373                | 517                  | 3                  | 0       | 9,323    | -9,329       | 23,938              |
| 26.095                | 94.253             | 120,285                   | 215,915                   | 16,091                   | 53,686               | 82,063               | 0                  | 882     | 0        | 0            | 1,197,573           |

Fig. 3.

---

## Author Comment (AC3) · 16 Feb 2021

(1) Equation 2 is incomplete. What is needed is the specification of the full IO table with the two virtual industries (garbage cleaning and reconstruction), which is only partially done in Table 2. In Section 4 and Table 2 it is explained rightly that garbage cleaning inter alia needs manpower, which is taken from the manpower row of existing industries. Amongst others, this is absent in (2).

REPLY: I will add the hybrid I-O table which includes garbage cleaning services and manpower sector.

(2) The cut-off method for resources described on p.3 is too rude. Equation 3 implies that structure of the use of resources from the rest of the economy by the new industry Cu is equal to the structure of the aggregate use of those resources in the rest of the economy Mi, where i is a summation vector with ones. This a highly unrealistic assumption because the new virtual industries take care of garbage cleaning and reconstruction, which both have cost structures that will be quite different from the cost structure of the rest of the Kochi economy that is dominated by services. In fact, Section 3, describing the details of the procedure for garbage cleaning and reconstruction activities, is contradictory to (3), but more realistic. The cut-off method for products also described on p.3 is imprecise and not convincing.

REPLY: We made an assumption that all industries will try to minimize the stress of each industry or the focal damage on one industry sector so they will simultaneously adjust the damage share among industries. There is no certain way to obtain the real damage data for each industry. Furthermore, the data is based on each scenario. For example, even though there are two flood scenarios in the same province but the locations of bank break are different. The characteristic of the flood damage will be different. Therefore, we made this assumption to forecast the prior outcome of each scenario.

(3) Equation 4 that should describe the structure of Cd is lacking. The most plausible solution would be to assume that the new industries do not deliver intermediate products (i.e. Cd is zero), but only deliver final products to fP, which goes at the cost of the final demand for products from the rest of the economy fM. As a consequence of the inconsistency of equation 5 (see Major details) Table 3 delivers nonsense.

REPLY: Cd is not lacking. The garbage cleaning service delivers their products to agriculture, public works and final demand.

(4) Equation 5 can only be used to establish the overall total of z12 (i.e. a small case as it is not a matrix, but a column). Instead of Table 3, the structure of the inputs to the

manpower sector should resemble the structure of the household consumption part of the final demand column (see Oosterhaven, Rethinking IO Analysis, Springer 2019, ch. 4). The procedure that describes from which sectors the manpower of garbage cleaning is taken should be specified independently. It is not related with the structure of household consumption demand.

REPLY: $Z21$ and $Z22$ are shifted from the value-added row for manpower sector row. The equation 5 shows how we constructs the manpower column $Z12$ and $Z22$. When the conventional I-O table is constructed, the summation of inter-industry row and column is equal as well as the summation of value-added and final demand. What is the meaning behind this? The money that is use for satisfying the final demand actually come from the value-added. Likewise, the manpower row is taken from the value-added and the manpower column is based on equation 4 and 5. Lastly, we put on the table 3 to give an example of how the manpower column, however you can see it clearly since the whole hybrid I-O table will be added to the appendix. Major details:

âAËŸ c P.1-2 & p.12-14. This article does not mention the direct los of output ′ of industries that have to partially or entirely close down due to the simulated flood nor does it mention the indirect impacts of these close down in the rest of the economy nor does it mention the cost in the rest of the economy of raising the money for the cleaning and reconstruction activities.

REPLY: Loss of opportunity or opportunity cost is not included in this analysis. It is our next step to implement the method since the opportunity cost requires more data and some methodologies to identify it.

The article, consequently, presents a far too optimistic view of the effects of the simulated flood for Kochi. If the authors believe their own results, they should advocate to have regular floods in Kochi. âAËŸ c P.7, l.12. The matrix ′ dimensions of equation 4 and 5 are inconsistent. In (4), first row, unity columns should be added behind the Z-matrices. In (4), second row, unity rows should be added in front of the Z-matrices.

Consequently, (5) is inconsistent at the matrix level. It is only correct for the overall matrix totals. âAËŸ c P.3, l.7 & elsewhere.

REPLY: It is certain that if your house, barn or car are damaged by flood, you need to maintain or fix it for your daily use. Some materials have to be purchased to ease the damage so the economy is benefited through an additional demand.

Write input coefficients instead of ′ technical coefficients, because you are dealing with an open economy in which: input coefficients = trade origin ratios * technical coefficients (see Oosterhaven, 2019, ch. 2). âAËŸ c P.8, l.17. As a consequence of the above misuse of the term technical, it ′ is incorrectly stated that input coefficients change only gradually due to technological advancement. In fact, they also change much faster due to spatial substitution. Not taking this into account leads to overestimation of the indirect damages of garbage cleaning in Section 4.3.

REPLY: I will change this term in accordance with the paper and add citation.

The authors might want to have a look at Oosterhaven and Többen (Spatial Econ An, 2017) for a solution to this problem. Minor details: âAËŸ c P.2, l.14. These costs are only part of the total impact. So, better ′ call them total direct impact. âAËŸ c P.3, l.5.

REPLY: I will change accordingly.

Write Miller and Blair, 2009. â ′ AËŸ c Figure ′ 2. The pre-process related to garbage cleaning, not to reconstruction. âAËŸ c The text ′ around Figure 2 fails to discuss how it differs from the quite comparable approaches of Hallegate and others (see also Koks et al, in Okuyama & Rose, Springer, 2019).

REPLY: I will add the full hybrid I-O table in the appendix.
* * *
| Hybrid I-O Table | Truck | Manpower | Garbage Cleaning | Agricultural | Livestock | Forestry | Fishery | Mining | Food and Beverage | Textile | Lumber, Wood Product and Furniture | Pulp, Paper and Paper Product | Print Plate Making and Book Binding | Chemical Product | Petroleum and Coal | Ceramic, Stone and Clay |
|---|---|---|---|---|---|---|---|---|---|---|---|---|---|---|---|---|
| Truck (hr) | -322,588 | | 322,588 | | | | | | | | | | | | | |
| Manpower (hr) | | -57,545 | 57,545 | | | | | | | | | | | | | |
| Garbage cleaning service (yen) | | | | 138 | | | | | | | | | | | | |
| Agricultural | | | | 2,995 | 946 | 19 | 0 | 0 | 12,694 | 99 | 0 | 322 | 2 | 0 | 1 | 27 |
| Livestock | | | | 3,684 | 1,209 | 1 | 0 | 0 | 7,330 | 20 | 0 | 0 | 0 | 0 | 0 | 0 |
| Forestry | | | | 24 | 0 | 2,535 | 9 | 1 | 43 | 0 | 6,144 | 0 | 2 | 0 | 0 | 0 |
| Fishery | | | | 0 | 0 | 0 | 1,376 | 0 | 16,192 | 0 | 0 | 0 | 0 | 0 | 0 | 0 |
| Mining | | | | 0 | 0 | 6 | 0 | 191 | 5 | 0 | 0 | 678 | 183 | 126 | 0 | 4,988 |
| Food/Beverage | | | | 490 | 2,661 | 93 | 4,808 | 0 | 18,548 | 4 | 5 | 159 | 14 | 0 | 0 | 19 |
| Textile | | | | 277 | 14 | 24 | 1,088 | 81 | 100 | 2,393 | 44 | 772 | 6 | 0 | 6 | 102 |
| Lumber, Wood Product and Furniture | | | | 5 | 12 | 22 | 105 | 18 | 122 | 14 | 1,093 | 54 | 6 | 0 | 7 | 65 |
| Pulp, Paper and Paper Product | | | | 3,561 | 261 | 32 | 83 | 0 | 1,459 | 144 | 144 | 19,475 | 160 | 0 | 45 | 456 |
| Chemical Product | | | | 6,858 | 339 | 19 | 577 | 305 | 833 | 2,261 | 239 | 2,700 | 2,365 | 74 | 1,517 | 490 |
| Petroleum and Coal | | | | 2,865 | 61 | 260 | 4,706 | 610 | 469 | 75 | 79 | 459 | 203 | 754 | 11 | 1,357 |
| Plastic and Rubber | | | | 1,604 | 65 | 146 | 814 | 112 | 1,820 | 285 | 155 | 2,303 | 203 | 2 | 1,772 | 265 |
| Ceramic, Stone and Clay | | | | 207 | 30 | 6 | 3 | 2 | 203 | 20 | 85 | 109 | 209 | 24 | 35 | 5,281 |
| Iron, Steel and Non-Ferrous | | | | 15 | 0 | 0 | 18 | 10 | 157 | 5 | 42 | 32 | 209 | 0 | 29 | 596 |
| Metal Product | | | | 96 | 6 | 12 | 102 | 182 | 755 | 43 | 122 | 77 | 113 | 0 | 14 | 262 |
| General Machinery | | | | 0 | 0 | 0 | 0 | 7 | 0 | 0 | 33 | 0 | 0 | 0 | 3 | 58 |
| Electrical Machinery | | | | 0 | 0 | 2 | 0 | 17 | 0 | 0 | 1 | 0 | 0 | 0 | 22 | 88 |
| Information/Communication Equipment | | | | 0 | 17 | 1 | 1 | 0 | 0 | 0 | 0 | 0 | 0 | 0 | 0 | 0 |
| Electronic Component | | | | 0 | 0 | 0 | 0 | 2 | 0 | 0 | 0 | 0 | 0 | 0 | 0 | 0 |
| Transport Machinery | | | | 1 | 1 | 1 | 65 | 16 | 1 | 0 | 2 | 1 | 0 | 0 | 0 | 5 |
| Precision Machinery | | | | 0 | 0 | 0 | 2,520 | 0 | 0 | 0 | 0 | 0 | 0 | 0 | 0 | 0 |
| Other Manufacturing | | | | 56 | 11 | 15 | 408 | 87 | 962 | 214 | 426 | 935 | 81 | 3 | 109 | 541 |
| Construction | | | | 1,289 | 61 | 39 | 111 | 104 | 145 | 79 | 28 | 794 | 78 | 20 | 53 | 814 |
| Public Work | | | | 0 | 0 | 0 | 0 | 0 | 0 | 0 | 0 | 0 | 0 | 0 | 0 | 0 |
| Electricity, Gas and Heat | | | | 831 | 285 | 62 | 288 | 904 | 1,383 | 291 | 212 | 5,561 | 385 | 40 | 154 | 4,677 |
| Water Supply | | | | 30 | 37 | 5 | 17 | 45 | 288 | 14 | 10 | 128 | 32 | 1 | 10 | 72 |
| Waste Treatment | | | | 0 | 14 | 0 | 0 | 8 | 64 | 1 | 10 | 12 | 14 | 0 | 0 | 115 |
| Trading | | | | 5,790 | 548 | 237 | 3,088 | 436 | 10,368 | 2,087 | 1,911 | 5,087 | 612 | 107 | 582 | 2,095 |
| Financial and Insurance | | | | 378 | 89 | 106 | 493 | 791 | 448 | 266 | 202 | 414 | 161 | 4 | 19 | 506 |
| Real Estate | | | | 73 | 109 | 9 | 46 | 79 | 237 | 59 | 41 | 124 | 55 | 4 | 26 | 176 |
| Transportation | 921 | | | 4,506 | 632 | 774 | 2,136 | 6,670 | 3,323 | 496 | 848 | 1,663 | 272 | 132 | 139 | 3,880 |
| Telecommunication | | | | 207 | 69 | 36 | 362 | 260 | 517 | 129 | 95 | 315 | 83 | 9 | 38 | 327 |
| Public Services | | | | 0 | 0 | 0 | 0 | 0 | 0 | 0 | 0 | 0 | 0 | 0 | 0 | 0 |
| Education and Research | | | | 12 | 2 | 32 | 99 | 95 | 514 | 294 | 41 | 936 | 646 | 3 | 158 | 1,578 |
| Medical and Healthcare | | | | 0 | 19 | 0 | 0 | 0 | 0 | 0 | 0 | 0 | 0 | 0 | 0 | 0 |
| Other Public Services | | | | 0 | 4 | 0 | 1,514 | 76 | 79 | 29 | 13 | 50 | 59 | 2 | 3 | 79 |
| Business Services | | | | 1,255 | 427 | 489 | 628 | 863 | 3,197 | 471 | 557 | 1,163 | 571 | 93 | 259 | 2,694 |
| Personal Services | | | | 0 | 8 | 5 | 60 | 4 | 552 | 2 | 2 | 6 | 0 | 0 | 0 | 4 |
| Office Supplies | | | | 21 | 12 | 30 | 65 | 48 | 64 | 19 | 8 | 38 | 8 | 1 | 1 | 51 |
| Unknown | | | | 1,284 | 185 | 251 | 825 | 141 | 303 | 34 | 106 | 77 | 69 | 32 | 9 | 412 |
| Manpower | | | 460 | 6,158 | 2,584 | 4,632 | 10,963 | 5,031 | 22,463 | 5,389 | 3,699 | 7,055 | 1,548 | 232 | 1,199 | 10,773 |

**Fig. 1.**
| Iron, Steel and Non-Ferrous | Metal Product | General Machinery | Electrical Machinery | Information Communication Equipment | Electronic Component | Transport Machinery | Precision Machinery | Other Manufacturing | Construction | Public Work | Electricity, Gas and Heat | Water Supply | Waste Treatment | Trading | Financial and Insurance | Real Estate | Transportation |
|---|---|---|---|---|---|---|---|---|---|---|---|---|---|---|---|---|---|
|  |  |  |  |  |  |  |  |  |  | 691 |  |  |  |  |  |  |  |
| 0 | 0 | 0 | 0 | 0 | 0 | 0 | 0 | 44 | 129 | 442 | 0 | 0 | 0 | 50 | 0 | 1 | 11 |
| 0 | 0 | 0 | 0 | 0 | 0 | 0 | 0 | 0 | 3 | 0 | 0 | 0 | 0 | 0 | 0 | 0 | 1 |
| 0 | 0 | 0 | 0 | 0 | 0 | 0 | 0 | 9 | 2 | 10 | 0 | 0 | 0 | 0 | 0 | 0 | 0 |
| 0 | 0 | 0 | 0 | 0 | 0 | 0 | 0 | 268 | 0 | 0 | 0 | 0 | 0 | 0 | 0 | 0 | 1 |
| 6,190 | 0 | 0 | 4 | 4 | 0 | 0 | 0 | 23 | 261 | 804 | 2,990 | 0 | 0 | 0 | 0 | 0 | 0 |
| 0 | 0 | 0 | 0 | 0 | 0 | 0 | 0 | 7 | 0 | 10 | 0 | 0 | 0 | 54 | 0 | 0 | 57 |
| 25 | 9 | 7 | 34 | 27 | 179 | 29 | 113 | 419 | 626 | 221 | 17 | 10 | 41 | 2,738 | 324 | 5 | 404 |
| 16 | 10 | 4 | 19 | 22 | 40 | 8 | 256 | 101 | 10,981 | 337 | 61 | 20 | 60 | 806 | 489 | 127 | 183 |
| 7 | 16 | 4 | 26 | 201 | 85 | 40 | 3 | 1,759 | 767 | 0 | 0 | 2 |  | 2,354 | 319 | 19 | 188 |
| 122 | 121 | 26 | 126 | 347 | 523 | 81 | 417 | 1,190 | 911 | 464 | 23 | 178 | 345 | 4 | 4 | 9 | 62 |
| 714 | 27 | 8 | 53 | 58 | 87 | 8 | 52 | 83 | 622 | 8,036 | 476 | 170 | 304 | 1,159 | 108 | 186 | 28,316 |
| 44 | 33 | 149 | 1,088 | 1,137 | 980 | 231 | 509 | 1,199 | 2,117 | 2,306 | 0 | 508 | 226 | 2,867 | 578 | 202 | 683 |
| 349 | 43 | 66 | 99 | 54 | 744 | 38 | 101 | 217 | 8,794 | 14,452 | 8 | 55 | 12 | 86 | 2 | 24 | 9 |
| 6,933 | 3,073 | 1,196 | 4,222 | 1,833 | 1,405 | 765 | 7,402 | 824 | 5,461 | 5,729 | 37 | 10 | 0 | 5 | 0 | 0 | 13 |
| 132 | 531 | 186 | 1,091 | 303 | 433 | 223 | 1,012 | 192 | 18,007 | 7,435 | 55 | 17 | 3 | 969 | 19 | 113 | 222 |
| 18 | 12 | 1,154 | 1,431 | 97 | 58 | 103 | 653 | 1 | 1,119 | 711 | 0 | 56 | 0 | 2 | 0 | 0 | 8 |
| 14 | 5 | 47 | 5,472 | 4 | 91 | 13 | 20 | 0 | 7 | 24 | 0 | 2 | 0 | 2 | 0 | 0 | 10 |
| 0 | 0 | 35 | 161 | 244 | 2 | 4 | 68 | 1 | 46 | 8 | 0 | 1 | 1 | 363 | 2 | 0 | 6 |
| 0 | 10 | 188 | 157 | 4,898 | 9,765 | 584 | 71 | 46 | 86 | 9 | 0 | 0 | 0 | 10 | 5 | 0 | 1 |
| 0 | 6 | 162 | 532 | 172 | 689 | 1,049 | 771 | 48 | 2,007 | 1,073 | 1 | 1 | 1 | 269 | 37 | 24 | 73 |
| 0 | 0 | 0 | 27 | 0 | 0 | 0 | 7,937 | 0 | 0 | 0 | 0 | 0 | 0 | 0 | 0 | 0 | 2,827 |
| 712 | 16 | 10 | 67 | 78 | 195 | 43 | 52 | 1,488 | 523 | 573 | 263 | 43 | 98 | 3,755 | 4,064 | 11 | 350 |
| 1,184 | 68 | 31 | 130 | 70 | 120 | 24 | 58 | 99 | 290 | 180 | 2,402 | 1,116 | 141 | 2,911 | 903 | 15,042 | 2,140 |
| 0 | 0 | 0 | 0 | 0 | 0 | 0 | 0 | 0 | 0 | 0 | 0 | 0 | 0 | 0 | 0 | 0 | 0 |
| 4,069 | 139 | 69 | 274 | 166 | 1,014 | 45 | 400 | 353 | 961 | 674 | 15,127 | 496 | 904 | 7,808 | 613 | 1,169 | 950 |
| 51 | 6 | 4 | 20 | 12 | 48 | 4 | 23 | 27 | 153 | 96 | 67 | 2,030 | 229 | 592 | 299 | 112 | 459 |
| 0 | 0 | 1 | 0 | 2 | 18 | 0 | 0 | 18 | 4 | 44 | 639 | 63 | 30 | 534 | 429 | 2 | 321 |
| 1,222 | 712 | 444 | 1,764 | 1,247 | 1,590 | 442 | 2,374 | 2,688 | 16,696 | 11,821 | 227 | 289 | 334 | 8,656 | 1,204 | 507 | 6,504 |
| 218 | 109 | 59 | 210 | 242 | 212 | 30 | 306 | 364 | 1,926 | 3,666 | 1,610 | 67 | 174 | 6,577 | 8,957 | 23,441 | 4,206 |
| 78 | 55 | 34 | 77 | 25 | 53 | 19 | 63 | 74 | 1,271 | 331 | 720 | 34 | 55 | 11,141 | 2,184 | 8,240 | 5,295 |
| 1,207 | 337 | 193 | 663 | 416 | 628 | 142 | 654 | 3,421 | 7,919 | 8,688 | 784 | 226 | 1,265 | 21,132 | 5,539 | 760 | 24,303 |
| 129 | 73 | 85 | 319 | 164 | 294 | 75 | 121 | 192 | 1,375 | 1,820 | 902 | 547 | 204 | 14,490 | 8,366 | 1,117 | 2,271 |
| 0 | 0 | 0 | 0 | 0 | 0 | 0 | 0 | 0 | 0 | 0 | 0 | 0 | 0 | 0 | 0 | 0 | 0 |
| 1,125 | 94 | 223 | 1,444 | 1,758 | 3,022 | 439 | 596 | 1,008 | 253 | 503 | 753 | 4 | 5 | 1,935 | 144 | 0 | 300 |
| 0 | 0 | 0 | 0 | 0 | 0 | 0 | 0 | 0 | 0 | 0 | 0 | 5 | 0 | 9 | 24 | 2 | 55 |
| 20 | 10 | 12 | 39 | 17 | 26 | 2 | 22 | 29 | 180 | 514 | 408 | 791 | 45 | 784 | 1,703 | 370 | 216 |
| 2,084 | 248 | 538 | 1,251 | 522 | 1,592 | 265 | 715 | 1,204 | 15,891 | 27,655 | 9,626 | 1,653 | 1,133 | 21,194 | 14,341 | 5,152 | 29,471 |
| 4 | 0 | 0 | 1 | 2 | 7 | 1 | 2 | 3 | 38 | 77 | 8 | 4 | 1 | 599 | 40 | 218 | 74 |
| 9 | 3 | 11 | 28 | 32 | 19 | 7 | 17 | 52 | 79 | 407 | 7 | 10 | 70 | 930 | 681 | 91 | 353 |
| 179 | 24 | 112 | 299 | 41 | 31 | 21 | 208 | 50 | 2,632 | 2,242 | 150 | 136 | 26 | 3,057 | 878 | 1,780 | 1,148 |
| 3,792 | 2,429 | 2,518 | 8,202 | 3,240 | 3,106 | 1,024 | 4,999 | 6,976 | 63,578 | 68,102 | 12,212 | 2,162 | 10,240 | 181,109 | 61,074 | 18,298 | 53,516 |

**Fig. 2.**

| Telecom munication | Public Services | Education and Research | Medical and Healthcare | Other Public Services | Business Services | Personal Services | Office Supplies | Unknown | Manpower | Final demand | Total Production |
|---|---|---|---|---|---|---|---|---|---|---|---|
| | | | | | | | | | | 0 | 0 |
| | | | | | | | | | | 0 | 0 |
| | | | | | | | | | | 552 | 1,381 |
| 0 | 4 | 9 | 767 | 47 | 2 | 2,799 | 0 | 0 | -17,004 | 82,252 | 86,658 |
| 0 | 0 | 78 | 116 | 0 | 0 | 769 | 0 | 0 | 5,077 | -4,678 | 13,610 |
| 0 | 1 | 0 | 30 | 0 | 0 | 150 | 0 | 0 | 3,691 | 6,568 | 19,219 |
| 0 | 1 | 0 | 225 | 0 | 0 | 1,323 | 0 | 0 | -7,029 | 39,589 | 51,946 |
| 0 | 1 | 3 | 0 | 0 | 0 | -3 | 0 | 2 | 4,291 | 1,053 | 21,800 |
| 0 | 74 | 108 | 3,527 | 36 | 1 | 26,058 | 0 | 29 | -26,413 | 92,743 | 123,092 |
| 104 | 1,621 | 60 | 1,638 | 1,676 | 330 | 897 | 105 | 17 | 6,735 | -4,944 | 18,374 |
| 286 | 343 | 270 | 1,517 | 590 | 214 | 1,078 | 0 | 4 | 6,667 | -7,030 | 19,002 |
| 2,145 | 95 | 522 | 1,266 | 147 | 455 | 723 | 2,735 | 48 | -4,682 | 23,887 | 58,967 |
| 256 | 262 | 378 | 58,365 | 86 | 620 | 1,299 | 102 | 307 | 78,404 | -152,789 | 10,820 |
| 154 | 2,513 | 753 | 1,768 | 219 | 508 | 1,997 | 0 | 782 | 59,639 | -118,561 | 2,148 |
| 240 | 447 | 297 | 869 | 251 | 2,033 | 603 | 478 | 193 | 24,792 | -47,554 | 7,052 |
| 1 | 59 | 295 | 308 | 17 | 263 | 284 | 35 | 185 | 734 | 22,000 | 55,548 |
| 14 | 40 | 4 | 610 | 8 | 148 | 98 | 6 | 461 | 14,555 | -12,035 | 43,930 |
| 52 | 972 | 22 | 143 | 92 | 218 | 579 | 2 | 100 | 29,117 | -54,667 | 9,357 |
| 1 | 57 | 0 | 0 | 0 | 1,018 | 2 | 0 | 0 | 1,544 | 273 | 8,419 |
| 0 | 3 | 0 | 0 | 0 | 1,677 | 2 | 0 | 0 | -13,605 | 40,453 | 34,371 |
| 17 | 1,685 | 0 | 4,435 | 0 | 901 | 1,753 | 165 | 0 | -4,278 | 13,396 | 19,035 |
| 107 | 362 | 66 | 2 | 0 | 1,957 | 2 | 187 | 0 | -5,435 | 16,818 | 29,898 |
| 62 | 474 | 128 | 76 | 4 | 1,426 | 87 | 0 | 33 | 4,563 | -7,163 | 6,698 |
| 0 | 1,760 | 10 | 0 | 0 | 12,259 | 4 | 0 | 0 | 2,330 | 3,336 | 33,010 |
| 2,994 | 3,283 | 2,446 | 2,056 | 1,691 | 1,528 | 1,780 | 897 | 40 | 15,414 | -20,053 | 28,265 |
| 1,088 | 6,746 | 2,026 | 2,364 | 172 | 405 | 1,300 | 0 | 0 | -57,549 | 194,142 | 181,218 |
| 0 | 0 | 0 | 0 | 0 | 0 | 0 | 0 | 0 | -101,957 | 290,009 | 188,052 |
| 1,029 | 1,627 | 3,697 | 4,805 | 127 | 963 | 4,123 | 0 | 232 | 30,122 | -26,105 | 70,924 |
| 379 | 776 | 1,470 | 2,612 | 83 | 122 | 2,342 | 0 | 78 | 4,272 | -402 | 16,653 |
| 495 | 7,403 | 749 | 1,205 | 1 | 43 | 3,707 | 0 | 94 | 10,347 | -5,857 | 20,530 |
| 1,803 | 3,274 | 2,019 | 24,961 | 1,584 | 4,336 | 22,630 | 1,486 | 364 | 36,284 | 210,888 | 401,298 |
| 748 | 10,750 | 170 | 2,586 | 1,738 | 1,690 | 1,797 | 0 | 115 | 23,589 | 73,002 | 172,436 |
| 2,249 | 312 | 663 | 8,570 | 749 | 1,186 | 3,276 | 0 | 933 | -9,999 | 276,599 | 315,325 |
| 3,281 | 7,435 | 3,916 | 6,611 | 1,188 | 2,504 | 8,372 | 313 | 1,959 | 27,836 | 24,590 | 191,753 |
| 24,514 | 6,759 | 2,762 | 6,308 | 2,508 | 8,560 | 5,779 | 0 | 1,013 | 30,457 | 6,420 | 130,071 |
| 0 | 0 | 0 | 0 | 0 | 0 | 0 | 0 | 5,247 | -67,994 | 323,311 | 260,564 |
| 2,295 | 34 | 426 | 1,881 | 0 | 407 | 144 | 0 | 922 | -9,615 | 161,242 | 175,752 |
| 82 | 5 | 0 | 12,426 | 0 | 2 | 11 | 0 | 77 | -167,494 | 600,057 | 445,280 |
| 227 | 1 | 121 | 1,347 | 0 | 356 | 2,920 | 0 | 48 | -4,150 | 25,706 | 33,672 |
| 15,286 | 12,932 | 7,192 | 14,097 | 2,775 | 14,596 | 7,919 | 0 | 1,262 | 160,719 | -232,478 | 151,502 |
| 1,670 | 168 | 169 | 10,103 | 116 | 230 | 4,018 | 0 | 72 | -93,298 | 330,294 | 255,264 |
| 264 | 813 | 445 | 1,009 | 169 | 211 | 427 | 0 | 4 | 0 | 0 | 6,514 |
| 894 | 149 | 2,466 | 1,608 | 192 | 1,373 | 517 | 3 | 0 | 9,323 | -9,329 | 23,938 |
| 26,095 | 94,253 | 120,285 | 215,915 | 16,091 | 53,686 | 82,063 | 0 | 882 | 0 | 0 | 1,197,573 |

**Fig. 3.**